# Cuneiform Tablets Micro-Surveying in an Optimized Photogrammetric Configuration

**Sara Antinozzi [1], Fausta Fiorillo [2,*] and Mirko Surdi [3]**

[1] Department of Civil Engineering, Università degli Studi di Salerno, 84084 Fisciano, Italy
[2] Department of Architecture, Built Environment and Construction Engineering, Politecnico di Milano, 20133 Milano, Italy
[3] Department of Languages and Cultures, Ghent University, 9000 Ghent, Belgium
[*] Correspondence: fausta.fiorillo@polimi.it

**Abstract:** In the current panorama of 3D digital documentation, the survey of tiny artifacts with micrometric details is strongly influenced by two factors: firstly, the still high cost of the instruments and technologies (active sensors) required to achieve the necessary level of accuracy and resolution; secondly, the needed professional skills for the macro-photogrammetric approach. In this context, this research aims to meet the demand for a digital survey and 3D representation of small objects with complex surfaces and sub-millimeter morphological characteristics using a low-cost configuration (passive sensors) for an image-based approach. The experiments concerned cuneiform tablets, which are challenging due to their morphological and geometrical characteristics. The digital replica of these unique artefacts can be helpful for their study and interpretation and many innovative applications: access and sharing, a collaborative interdisciplinary study among several experts, experimentation with machine learning for automatic character recognition, and linguistic studies. The micrometric surveying system described proves to be an efficient and reliable solution for cuneiform tablet digitization and documentation.

**Keywords:** tiny artifacts; digitized heritage; micrometric details; handheld microscope; macro-photogrammetry; cuneiform tablet

## 1. Introduction

Digital documentation techniques for cultural heritage (CH) are nowadays well established and well- defined for detailed drawing sizes and scales, but still not conveniently formalized and codified for tiny objects. In fact, small artifacts represent a complex challenge due to the level of detail and accuracy required.

The tiny object 3D reconstruction is widely used in many fields, such as mechanics or medical sciences [1], thanks to the technologies that perform very well in terms of the specific size and representation scale requirements. Advanced research in micro- and nano-metrology now challenges its limits even further as the critical dimensions shrink and the geometric complexity of objects increases [2].

However, several examples of micrometric survey applications using high-resolution 3D scanning can also be found in the CH field [3]. Unfortunately, these technologies are often beyond the reach of most museums and cultural institutions due to the considerable hardware and software costs [3] and the level of expertise required by operators [4]. Consequently, the current market demand for practical and low-cost solutions directs interest towards cheaper alternatives, such as macro-photogrammetry. Actually, image-based approaches for the detailed digital documentation of even tiny artifacts have become widespread in CH applications [5–8].

Such image-based approaches have the great advantage of offering the possibility of obtaining the three-dimensional coordinates of an object and, at the same time, the exact

corresponding radiometric information from two-dimensional digital images in an accurate, reliable, flexible, and, often, inexpensive way. In the current panorama of 3D digital documentation systems, two factors strongly influence the survey of tiny artifacts with micrometric detail. The first is the still very high cost of the instruments and technologies (active sensors) required to achieve the level of accuracy and resolution[1]. The second is the specialized skills needed in macro-photography and micro-photogrammetry, which is certainly characterized by lower costs (but still not low, due to the cost of the macro-optics and some indispensable accessories). In addition, acquiring the photographic dataset for tiny objects currently represents a repetitive and time-consuming process [9].

The presented research focuses on configuring an optimized acquisition system for small items based on low-cost passive sensors with high magnification requirements, such as USB portable microscopes[2] (whose price starts at a few hundred euros). The recent use of photographs taken with digital microscopes as a dataset for photogrammetric processing has revealed a non-negligible potential for modeling small objects, especially when compared to significantly more expensive tools [10]. Therefore, the authors have previously tested the photogrammetric capability of a USB portable microscope [11,12] and have now applied it to larger data samples to confirm its effectiveness.

Although the effort is to employ more affordable hardware for micro-scale surveys, the research also aims to find an optimal geometry to suit the unusual photogrammetric instrumentation and to facilitate operator work. Indeed, some of the operational difficulties experienced by the authors recently [12] show that instrumentation that was not born in the purely photogrammetric field can be effective but that it needs to be adapted and that it also needs to deal with problems naturally related to macro-photogrammetry [13,14], such as the lack of a wide dynamic range, the narrow field of view, and the poor depth of field.

The choice of cuneiform tablets, a particularly complex case study in terms of both geometry and texture, allows for stressing the proposed acquisition system concering the problems just mentioned and allows the validation of the methodology. The success of the tests on this type of artifact with many tiny details can allow us to ensure that an accurate digital clone of the original cuneiform tablets can be quickly obtained at a low cost.

The digital replica can be helpful for many reasons and has many innovative applications: (i) it allows the virtual manipulation of these delicate findings; (ii) it allows the access and the sharing of artifact documentary information; (iii) it facilitates a collaborative interdisciplinary study among several experts; and (iv) it allows experimentation with machine learning for automatic character recognition and linguistic studies [15–17]. The digitization of cuneiform tablets in 3D has many positive implications for both research and 'edutainment' [18]. For example, the 3D models can also be utilized to establish digital libraries [19,20] for educational and scientific reasons, thereby facilitating an interdisciplinary study among various experts and providing new information and new representations of the tablets that were not available with the traditional documentary methodologies.

The present article is organized as follows. In Section 2, various methods for documenting and representing cuneiform tablets have been described and compared in order to provide a better understanding of the methodologies already in use and their advantages and disadvantages from an Assyriological perspective. Section 3 contains the test on the use of a digital microscope for accurate micro-photogrammetric reconstructions, the research to design an optimized data acquisition set, and the method validation using a benchmark. Section 4 describes the calibration procedure for the optical system, which enabled rigorous results to be obtained. Finally, Section 5 reports the digitization campaign of some cuneiform tablets belonging to the collection of Ghent University and the resulting models.

## 2. Cuneiform Tablets Digital Documentation: Aims and Purposes

### 2.1. Overall Characterization

Cuneiform tablets are among the oldest written documents of human history, dating back to 3300 BC. For over three millennia, clay tablets were by far the most common media for conveying information in the ancient Near East. Among the many shapes of the extant clay documents [21] (pp. 9–10), the most frequent are square or rectangular pillow-shaped clay artifacts, generally covered with cuneiform signs impressed on the wet clay with a stylus. The dimensions of the tablets can vary according to the genre, chronology, and provenance of the texts. Cuneiform tablets range from ca. 1.5 × 1.5 cm to 36 × 33 cm in size [22] (p. 75), but most of them usually fit comfortably in one's hand (Figure 1).

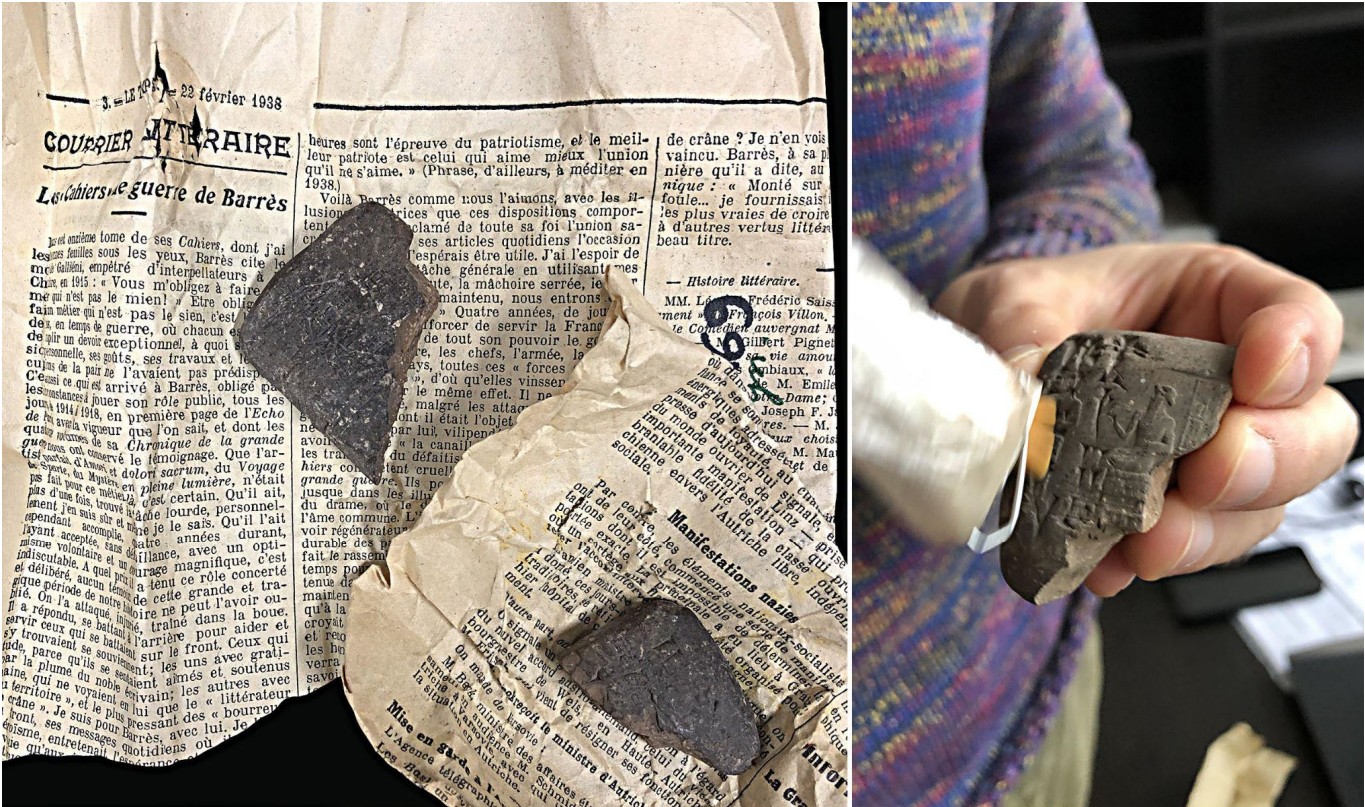

**Figure 1.** Some fragments of cuneiform tablets belonging to Ghent University.

According to an estimate by the Assyriologist M.P. Streck [23], more than 530,000 cuneiform objects (including cuneiform tablets) have been unearthed and are currently stored in different collections worldwide. Unfortunately, less than one-quarter of them are published. However, more tablets are being unearthed yearly by legal and illegal excavations.

Therefore, despite our access to a wealth of written sources of ancient history, we do not have an accurate survey of what is actually available, what is buried in museums, and what is still resting in the mounds of the Middle East. The enormous number of administrative texts that have so far survived can be accounted for by the simple yet essential fact that clay tablets can survive fire and other agents of deterioration much better than other materials used elsewhere or later (such as papyrus, parchment, and paper). Furthermore, over the course of thousands of years, thanks to its versatility, the cuneiform script has been used for nearly fifteen different languages in the ancient Near East area: from Iran to the Mediterranean, from Anatolia to Egypt [24] (p. 40).

In summary, cuneiform tablets: (i) are three-dimensional artifacts in clay; (ii) exhibit a three-dimensional script, characterized by wedges measuring even a few millimeters;

(iii) have different sizes, but most commonly, they fit in one's hand; (iv) are available in hundreds of thousands of exemplars; and (v) are nowadays scattered in collections all over the world, and more exemplars are unearthed every year all over the Middle East.

Documenting and publishing these precious clay artifacts have always been challenging due to the above characteristics. Based on the technologies available at the time, Assyriologists attempted to document cuneiform tablets in the most convenient and accurate manner possible.

*2.2. Acquisition and Publication: Limits of the Standard Methodologies*

For the acquisition and publication of cuneiform tablets, the most common methods are hand tracing copies, photographs, flatbed scans, and, more recently, 2D+ height maps based on polynomial texture map (PTM) technology and 3D modeling by photogrammetry and structured light scanning.

The salient features of the procedures for the study and visualization of these complex surfaces are briefly illustrated below. Furthermore, special attention is paid to the advantages and disadvantages of each methodology from an Assyriological perspective. By doing so, we seek to (i) contextualize the technique proposed in this article within the group of methodologies already in use and (ii) evaluate the effectiveness of micro-photogrammetry for cuneiform tablets compared to the other techniques.

2.2.1. Hand Tracing: Ink on Paper and Vector-Based Techniques

Hand copies and photographs represent the two very first and oldest methods used for publication in the early days of Assyriology, and both are still used today. In order to best read a cuneiform tablet, that is, to identify the signs correctly, it is necessary to vary the light source. Thus, the tablet is usually held in one's hand and rotated in different directions and angles to find the optimal illumination that will reveal the signs. Consequently, it is easy to understand why, even in the presence of the original artifact, many difficulties exist in reading the text, possibly due to the tablet's morphology and its current state of preservation.

Despite these difficulties, Assyriologists have made many cuneiform texts available to the scientific community. Traditionally, cuneiform tablets are copied using pencils, calipers, and graph paper. The skills of the person who is copying the tablet, as well as the amount of time available in the museums, collections, and archaeological sites determine the quality and accuracy of the hand copies.In some cases, the hand copies may appear sketchy and may represent only the cuneiform signs preserved without any other detail (Figure 2a).

In the best cases, Assyriologists have attempted to reproduce every detail of the tablet on paper, including the cuneiform signs and any fractures or abrasions that may have rendered the original text unreadable (Figure 2b).

Even though this method may seem outdated today, autographed copies of tablets are still used for specialized scientific publications. Despite the best efforts of the copyist, errors can occur; in fact, it is essential that the Assyriologist first reads the sign, then interprets it, and finally copies it on paper.

Every step in this chain of actions can be affected by human error, e.g., misreading the original sign on the tablet, misinterpreting and/or confusing it with another similar sign, and finally miscopying the sign on the paper.

Due to the aforementioned reasons, it is not uncommon for scholars to collate the tablet, i.e., to verify the accuracy of the hand copy through an autoptic inspection of the original tablet.Hand copies have numerous advantages since they require, at the very least, a pencil and some paper, which means that they can be produced anywhere at any time. Once the hand copy is ready, it is usually scanned and reproduced in the publication. In contrast, the disadvantages are multiple and include the time required for copying, the time available for execution, the copyist skills, and the non-objectivity of the copy.

Nowadays, digital tracings of cuneiform tablets are a more accurate and efficient method of copying cuneiform texts. The chain of action is identical to the ink on paper hand copies (see above), but now cuneiform signs, fractures, and abrasions are traced on graphic devices using vector-based graphic software (such as Adobe Illustrator, Inkscape, etc.). Born-digital hand copies can be modified and processed multiple times.

One of the greatest advantages of born-digital tracing is the possibility to create autographs on a layer overlapped with the photo of a cuneiform text. As with ink and paper hand copies, this method has the same disadvantages: its time-consuming nature and lack of objectivity. In addition, graphic tablets can be expensive and are not readily available everywhere at any time, unlike ink and paper.

### 2.2.2. Digital Photography

Photography can overcome the drawbacks of hand copies, namely their potential non-objectivity. Unfortunately, it is challenging to capture, with two-dimensional images, an irregular three-dimensional tablet with three-dimensional wedges (Figure 3).

Other factors may affect the reading of a cuneiform tablet on a photograph: (i) different shades of colors on the surface as a result of (accidental) firings; (ii) stains (such as iron oxides black spots); and (iii) brighter areas of salt efflorescence [25].

Since the 1970s, the technique of covering cuneiform tablets with a thin layer of ammonium chloride (NH4Cl) to enhance the contrast in photographs and make uniform the areas of discoloration has become increasingly popular [26] (pp. 14, 32 fn. 21), [27].

Despite the use of ammonium chloride, it is still difficult to accurately represent 3D objects with a photograph (Figure 4).

With the introduction of digital photography, photographs can be digitally modified to improve the readability of cuneiform tablets [28,29].

Unfortunately, digital photography still suffers from the same limitations as analog photography, namely the inability to represent a complex three-dimensional object in its totality in two dimensions.

In addition to the costs of the devices and, potentially, of the service, one should also consider taking the photos in a controlled environment, which is not always possible. Cameras, however, can be easily transported anywhere. In the case of tablets coated with a thin layer of ammonium chloride, good results also rely on an additional operator who should work carefully to handle the toxic substance.

### 2.2.3. Flatbed Scanning

Flatbed scanners proved to be an effective, easy-to-use, and cheap alternative to photography. The Cuneiform Digital Library Initiative (CDLI) (http://cdli.ucla.edu/, accessed on 7 October 2022) has developed clear guidelines for scanning a cuneiform tablet: each side of the tablet is scanned and then edited and assembled into a single file [30].

The images, just as in photography, offer a clear, objective view of the artifact, but again, the convex, irregular surface of the tablet causes difficulty in illuminating each of the cuneiform wedges, and some areas can be completely blurred or without the necessary contrast (Figure 2c).

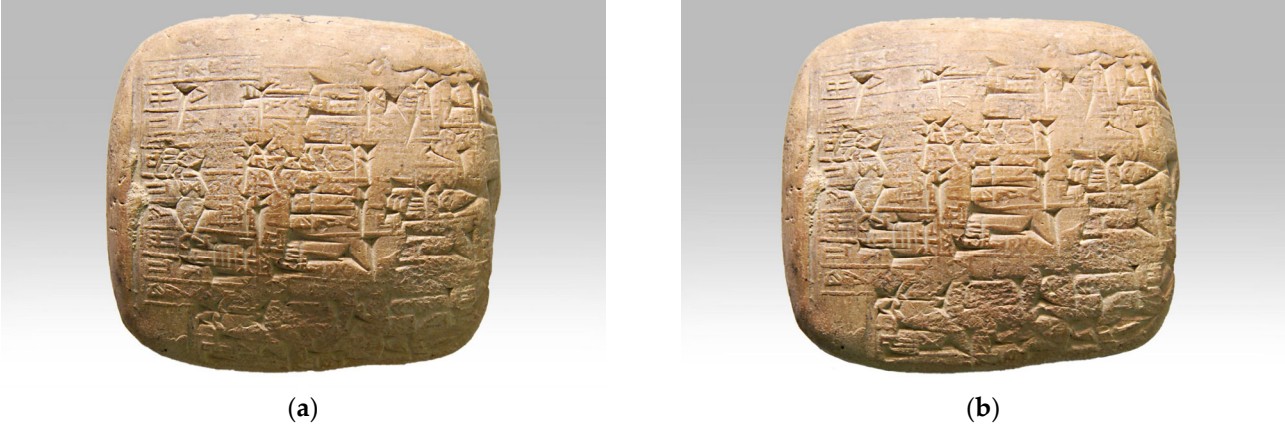

**Figure 2.** Tablet Ashm. 1911-237a-b from Umma (mod. Tell Jokha), datable to the Ur III period (2112-2004 BC): (**a**) copy adapted with permission from [31]. 1912, Langdon, S.H.; (**b**) copy adapted with permission from [32]. 1996, Grégoire, J.-P.; (**c**) flatbed scan from http://cdli.ucla.edu/ CDLI no. P142747 (accessed on 7 October 2022).

(**a**)                                    (**b**)

**Figure 3.** The same tablet (LW21.CUN.133) belonging to Ghent University required two different pictures with two different lighting sources to make all signs readable: (**a**) highly grazing light that accentuates the contrasts but darkens the lower part; (**b**) slightly grazing light that leaves the wedges at the bottom legible.

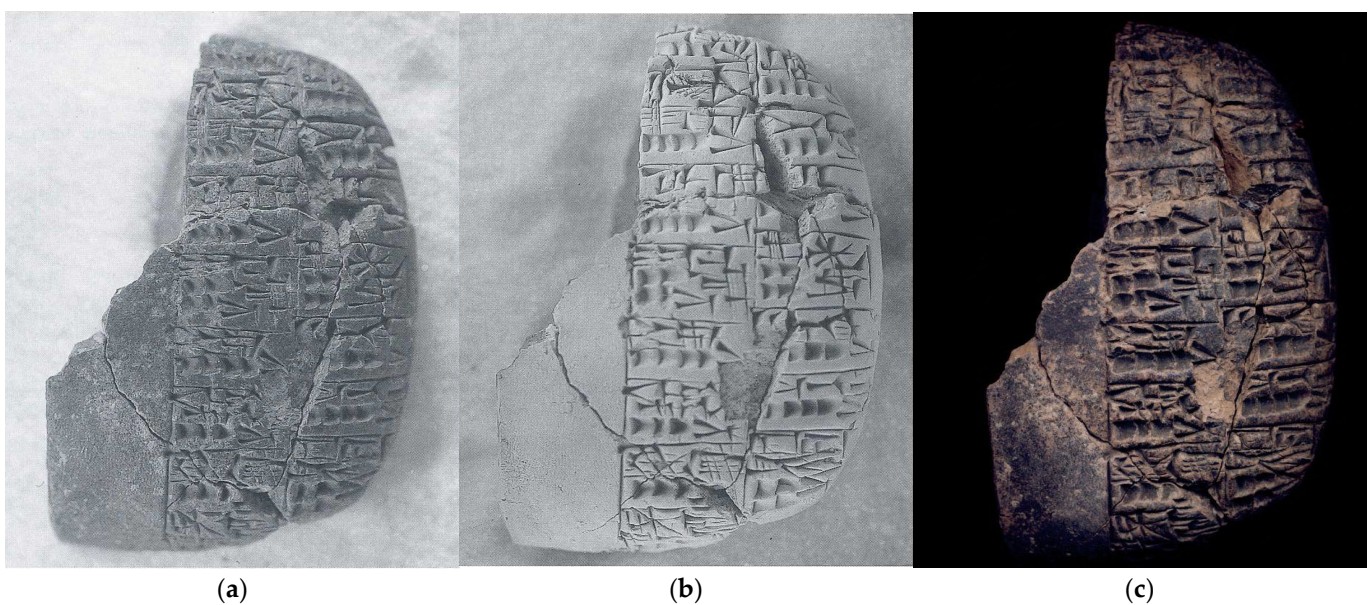

| (**a**) | (**b**) | (**c**) |

**Figure 4.** Cuneiform tablet from Tell Beydar (NMSDeZ Bey 95-001: verso): (**a**) photograph of the untreated tablet; (**b**) photograph of the tablet treated with ammonium chloride, reprinted with permission from [29]. 1997, Vandecasteele, C.; (**c**) tablet captured with a flatbed scanner, from https://cdli.ucla.edu/ CDLI no. P227324 (accessed on 7 October 2022).

### 2.2.4. RTI/PTM and Portable Light Dome

PTM (polynomial texture map) or RTI (reflectance transformation imaging) models are generated from multiple photographs of an object taken from a stationary camera. The light that is projected into each photograph comes from different known directions [33–37]. Based on the lighting information in the images, a mathematical model of the surface is produced, which enables the user to inspect the object by re-lighting it interactively. Additional effects and filters can be applied in order to improve the readability of the cuneiform writing.

The advantages of the RTI technique are many: good results can be obtained even with technical equipment suitable for conventional digital photography. The post-production and data processing have times that are also compatible with the study of archaeological artifacts in the field.

Furthermore, thanks to the model size, the files can be easily uploaded and downloaded and viewed with relative ease by an average user. The elaboration phase can produce models with different qualitative degrees of definition according to the needs. However, the RTI mapping does not produce an integrable three-dimensional model; therefore, an acquisition must be performed for each side of the tablet.

The 2D+ model produced is excellent for a detailed study of artifacts in a poor state of conservation but less suitable for a museum exposition of the tablet. Although 2D+ models are among the best methodologies for studying cuneiform tablets, some problems may arise as a result of their morphology (Figure 5).

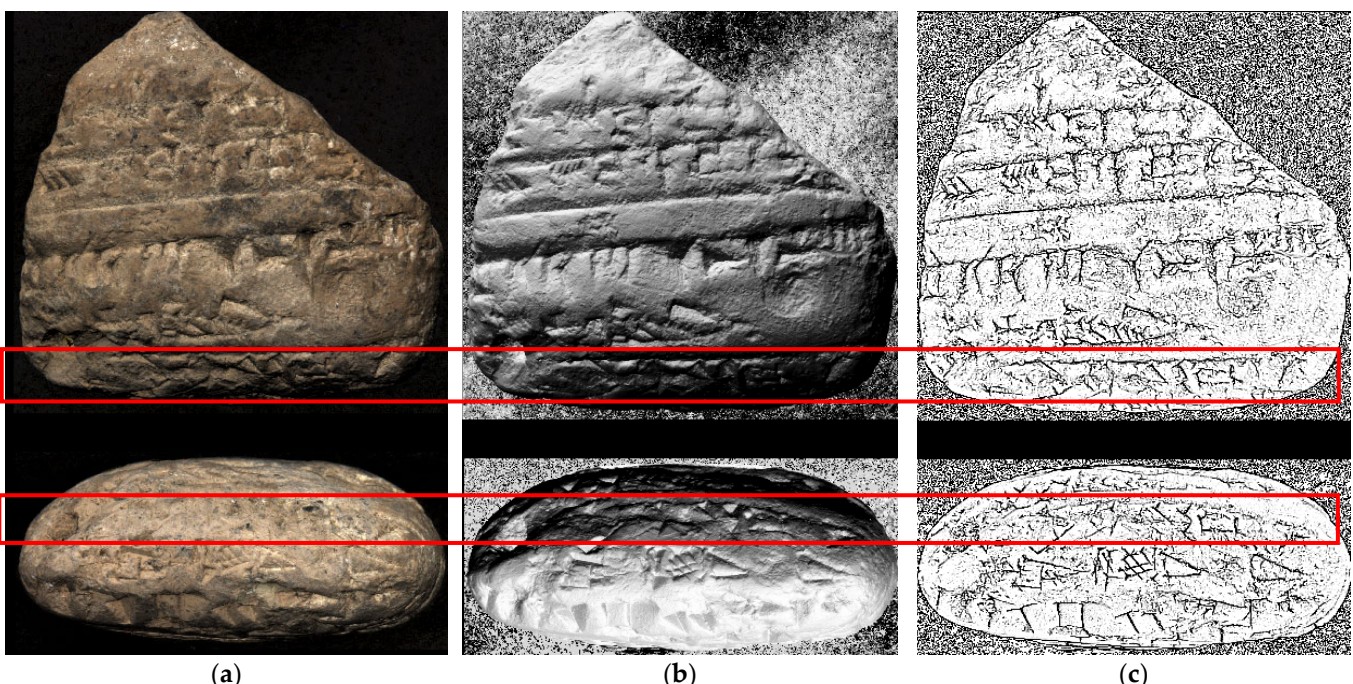

**Figure 5.** 2D+ models from PDL of tablet LW21.CUN.166 (Ghent University): (**a**) color visualization; (**b**) shaded visualization; (**c**) "Sketch 1" visualization. Neither the verso nor the bottom edge of the 2D+ model clearly exhibit the cuneiform text highlighted in red.

### 2.2.5. 3D Modeling: Photogrammetry and Structured Light Scanning

Thanks to active or passive sensor technologies, it is currently possible to obtain a faithful 3D copy of the physical model at a resolution of a tenth of a millimeter or even a micron. One of the significant advantages of photogrammetry is the possibility of simultaneously detecting geometry and color with photographic quality. Compared to the active sensors and the other previously illustrated techniques, it generally requires higher data acquisition and processing times.

Among the technologies based on active sensors, structured light scanners (SLS) ensure an excellent compromise between data resolution and survey speed. Often, a lacking aspect is the poor texture quality, and so, it can be less attractive for museum exhibitions. The costs of these instruments increase as the accuracy and resolution required increase.

The comparison between the active and passive sensors shows that the SLS benefits are faster data acquisition and processing times and the cleaning of the mesh surface.

However, the expenses are significantly higher, and the texture quality is not always on par with the geometry quality. Photogrammetry, on the other hand, has the advantage of high-quality texture and a lower cost.

However, it is characterized by much longer times, both for the acquisition of data and for their processing. In this framework, photogrammetry with a digital microscope (the solution described in the present research) represents a cost-effective approach because the microscope and the related equipment can be significantly less expensive than a professional camera and photographic kit. However, the texture has good quality, but not at the level of a professional photographic camera.

Table 1 collects and compares the main characteristics of all the systems analyzed.

**Table 1.** Gathering and comparison of the key attributes of all the systems examined. For clarification purposes: [a] exactness: how close the final model is to a true representation of the item; [b] acquisition time: the average time required to create a high-quality copy of small to medium-sized tablets (such as those illustrated in the present article); [c] lightening: the possibility to change/manage lighting effects at will and interactively on the result; [d] accessibility of technology: how widespread and popular the instrumentation indicated is.

| | 2D | | | | 2D+ | 3D | | |
|---|---|---|---|---|---|---|---|---|
| | Hand Tracing | | Raster | | PTM | Photogrammetry | | SLS |
| | Ink on Paper | Vector-Based | Flatbed Scanner | Photography | Portable Light Dome | Reflex & Macro Lens | USB Microscope | Scan in a Box |
| **Objectivity** | no | no | yes | yes | yes | yes | yes | yes |
| **Exactness** [a] | low-med | low-med | high | high | high | high | high | high |
| **Acquisition time** [b] **(1 tablet, 6 sides)** | ±24 h | ±24 h | ±10 min | ≤10 min | ±30 min | ±1 h | ±3 h | ±30 min |
| **Time of data elaboration** | ±1 min | immediate | ±5 min | ± 5 min | ≤20 min | ±3 h | ±3 h | ±5 min |
| **Required skills** | med-high | med-high | low-med | med-high | med-high | high | high | med-high |
| **Level of Detail** | low | low | med-high | med-high | high | high | high | high |
| **Accuracy** | low-med | low-med | high | high | high | high | high | very high |
| **Text readability** | high | high | low-med | medium | high | high | high | high |
| **Lightening** [c] | no | no | no | no | yes | yes | yes | yes |
| **Texture data** | no | no | no | no | yes | yes | yes | yes |
| **Accessibility of technology** [d] | high | medium | medium | high | med-high | medium | low-med | low |
| **Dissemination** | high | high | high | high | medium | low | low | low |
| **Data size manageability** | high | high | high | medium | med-high | low | low | low |
| **Equipment cost** | very low | med-high | medium | med-high | med-high | high | medium | high |

In this context, the digital survey, in general, and the resulting textured 3D models are the best option from an Assyriological perspective. First, the 3D tablets can be objectively analyzed, i.e., they are not filtered by human work like with the hand copies. The edges and marginal areas that are often not clearly visible in photographs can be visualized by rotating the model. A light simulator on the models can be used to simulate different directions of illumination so that the signs on the tablets can be read more clearly.

Post-processing software can also be used to enhance the visibility of the morphological details on the 3D model. For example, an effect similar to that of the ammonium chloride photographs can be achieved by applying specific filters (such as MeshLab's "Radiance Scaling" shader) to the 3D models to make the discoloration uniform and the wedges more prominent and thus improve their legibility (Figure 6).

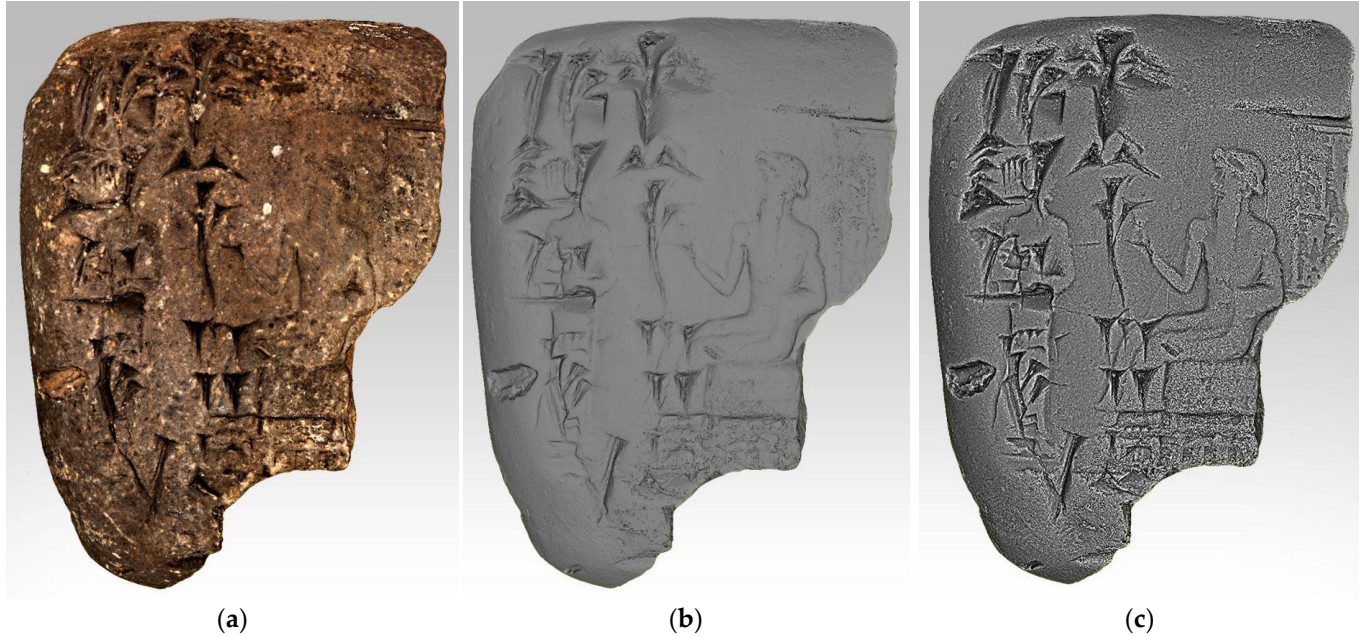

　　　　　　(**a**)　　　　　　　　　　　　　　　　　(**b**)　　　　　　　　　　　　　　　　(**c**)

**Figure 6.** Tablet LW21.CUN.159 belonging to Ghent University: (**a**) model textured; (**b**) mesh model; (**c**) mesh model with Radiance Scaling plugin by Meshlab. Mesh model from DSLR Camera Nikon D800E combined with AF-S Micro NIKKOR 60 mm f/2.8 G ED.

Furthermore, there are two other advantages to using 3D models: the possibility of sharing online digital content and, eventually, the possibility of virtually joining one or more fragments of the same tablet together that do not necessarily belong to the same collection and might not be in the same place [38] (pp. 272–273). In the case of the 3D models, two fragments can be joined together virtually without moving the real fragments from the original collection (Figure 7).

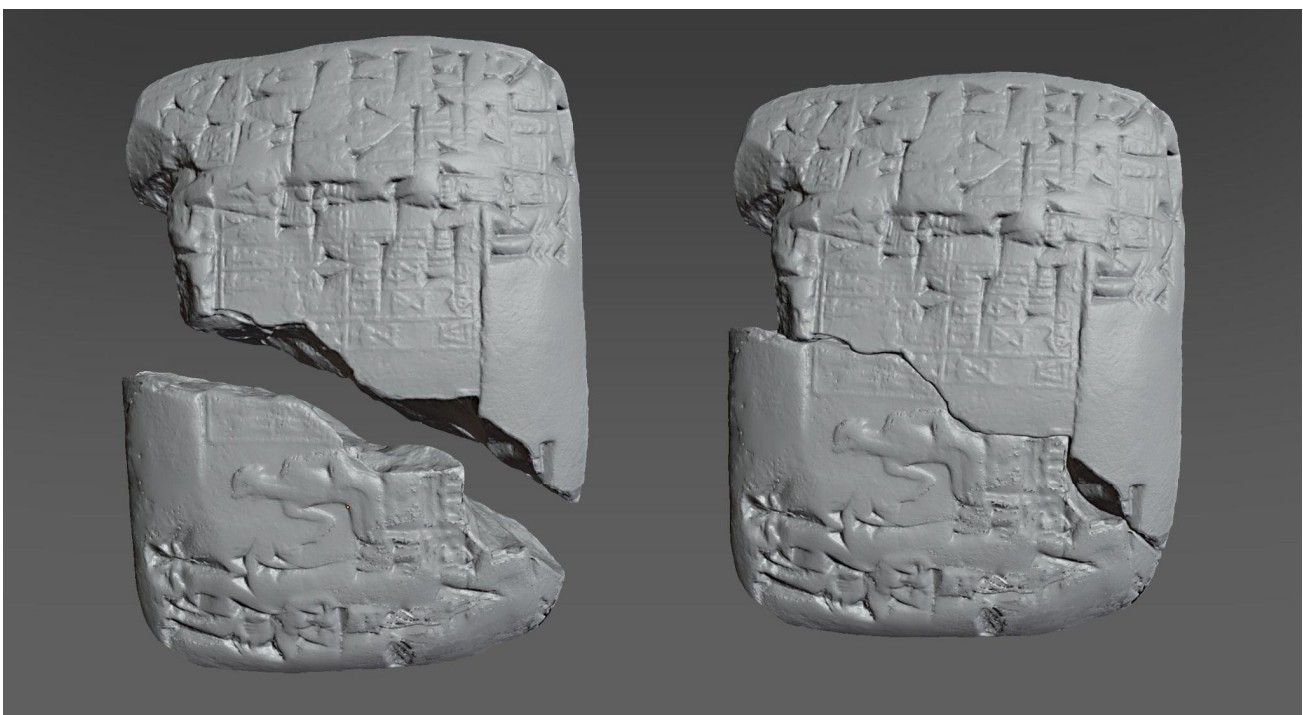

**Figure 7.** Tablet LW21.CUN.159 (verso, bottom piece) and LW21.CUN.160 (verso, upper piece) belonging to the Ghent University and digitally joined. Mesh model from DSLR Camera Nikon D800E combined with AF-S Micro NIKKOR 60 mm f/2.8 G ED.

## 3. Towards the Identification of an Optimized Acquisition System

*3.1. USB Microscopes as a New Micro-Photogrammetric Tool*

A digital microscope (Figure 8) is essentially the same as a conventional optical microscope without the eyepiece but with a CCD camera instead [39]. Therefore, USB handheld digital microscopes are more flexible, easy to use, and less expensive than traditional models.

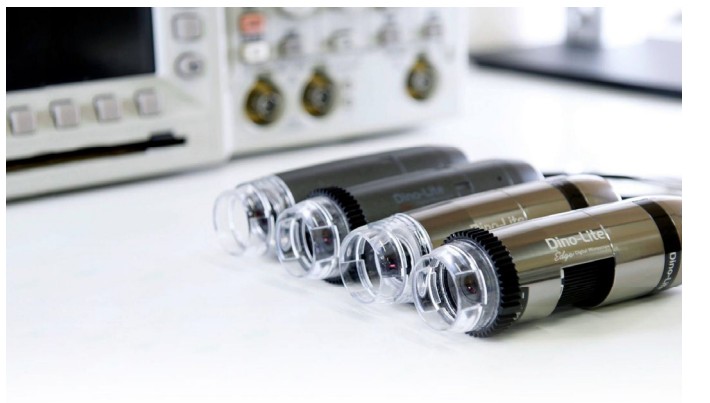

(**a**)

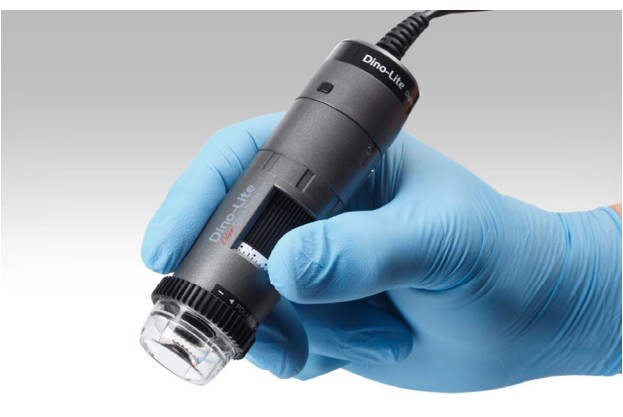

(**b**)

**Figure 8.** Dino-Lite USB digital microscopes: (**a**) different models; (**b**) handheld configuration.

Born for the inspection, documentation, and digital metrology analysis and already widespread in the manufacturing industry for quality control and in the medical field, digital microscopes have also been used for some applications of a micro-photogrammetric survey, even reaching a precision of about a tenth of a mm [40].

Therefore, integrating these tools into a structured photogrammetric workflow seemed promising. Their main drawback is that they were not designed from the start for



photogrammetric applications and have therefore been adapted to increase the accuracy of the results, such as through the design of a particular calibrator and of an automatic system that accelerates the data acquisition.

Furthermore, for the same reason, no technical calibration characteristics of the lens system are widespread so far.

### 3.2. Initial Experiments

The authors used a physical replica of a cuneiform tablet with wedges, which were as detailed, challenging, and complex as the original (Figure 9), to perform the first tests to evaluate the photogrammetric use of a digital microscope.

The clone is a physical 3D printing of the original tablet no. 724 realized by the +LAB (www.piulab.it, accessed on 7 October 2022) and digitally surveyed by the 3D Survey Group—Politecnico di Milano[3].

The text concerns the administrative information for fish deliveries. This case study was chosen as a test case for the micro-survey system because it was the smallest and most complex. Indeed, the tablet is about 20 × 22 × 8 mm in size, and its distinctive wedge-shaped impressions have a depth of 1–3 mm.

The early research has yielded positive and promising results [11]. The next step was to deepen some theoretical aspects and streamline the acquisition procedures, ensuring a more rigorous workflow [12].

The first aspect on which our research focused was the design of an ad hoc acquisition setup for this micro-scale of size and detail. Therefore, during several acquisition campaigns, the following were studied and tested: a calibrated support to arrange the object and simultaneously provide a metric reference; a rotating base to simplify and speed up acquisitions; and a specific lighting setting.

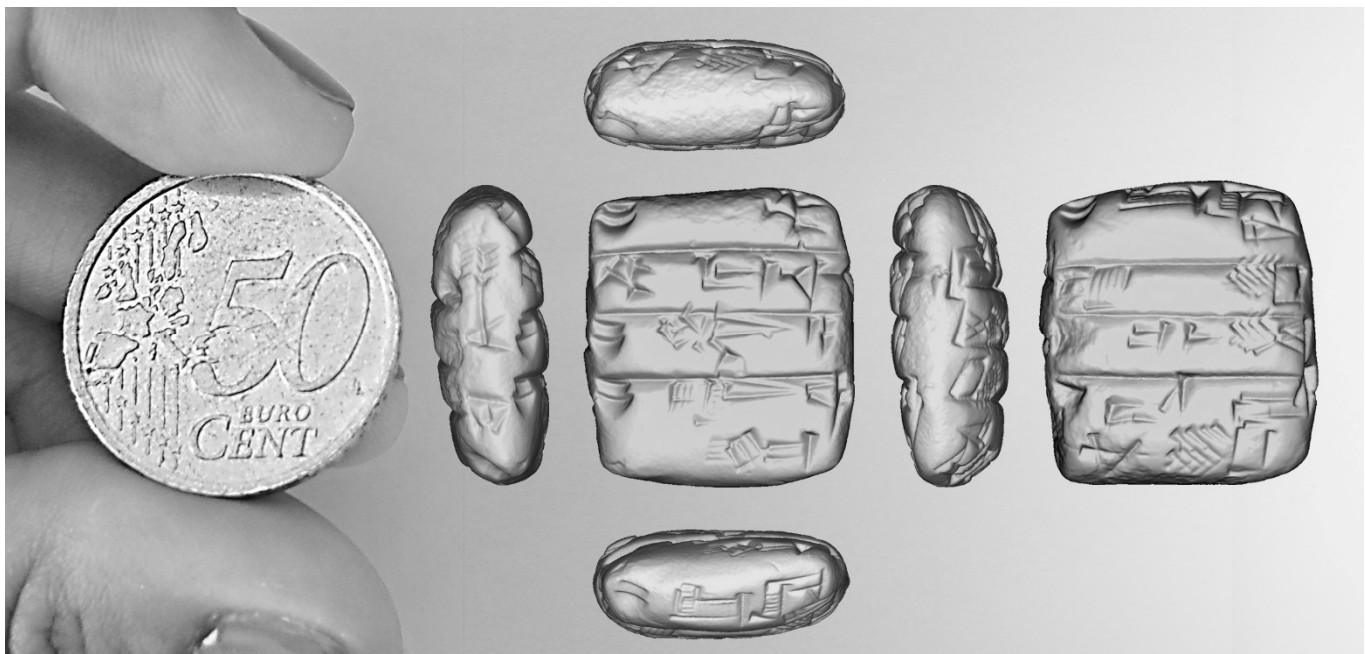

**Figure 9.** Scale comparison of the original tablet no. 724 from the private collection Durac-Donner (Genève). Mesh model from Scan in a Box 3D scanner.

The first configuration used (Figure 10) for data acquisition with the microscope consisted of a calibrated plate, two LED illumination rings (one around the object and the other above it), and the Dino-Lite RK-06A stand [41] to house the microscope with a vertical axis (standard configuration).

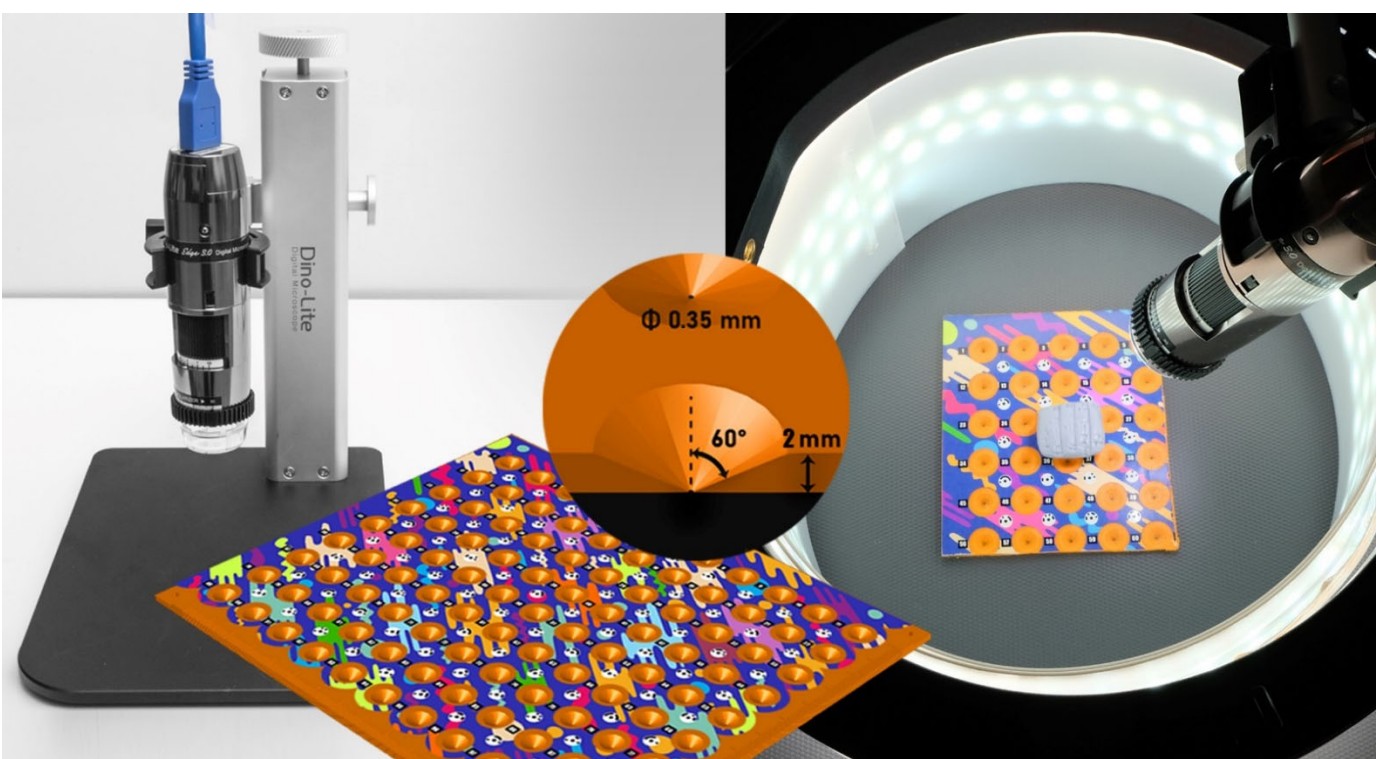

**Figure 10.** The raw configuration components: on the left, the standard stand to house the microscope with a vertical axis; in the middle, the plate calibrator with a section detail of a truncated cone hole; on the right, two illumination rings combined with the microscope asset and the cuneiform tablet on the calibrated plate.

The plate, printed with the PLA filament, was designed with an orthogonal pattern of 99 truncated conical holes, with a countersink angle of 60° and a base diameter of less than 0.35 mm. The calibrator accuracy was estimated to be 0.1 mm (marker accuracy), based on the 3D print settings and the compliance of the physical hole locations with the design file.

In this configuration, the digital microscope is stationary, housed on the vertical bracket at a fixed angle. Instead, the calibrated plate is manually slid for the image acquisition of the tablet surface.

Several problems were encountered: long acquisition times, which were also due to the difficulty in keeping the plate integral with the object during sliding; overestimated target size on the plate; and the impossibility of rigorously aligning the two faces of the tablet.

Indeed, the two sets of images, one for the recto and the other for the verso, do not have enough common points to be merged into a single model. It would have been necessary to survey the thin edge with severe depth-of-field problems.

To solve the problems encountered during the first test, two configurations were designed for the acquisitions. The first setup improved the one previously described with the microscope vertical axis on a customized tripod and the horizontal support plate for the cuneiform tablet (Figure 11).

A new alternative solution consists of a clamp to house the tablet with a vertical axis and a tripod to position the microscope with a horizontal axis (Figure 12). On the clamp support, a pattern with the marker is glued; therefore, even in this second mode, it is possible to frame the metric references and the object simultaneously.

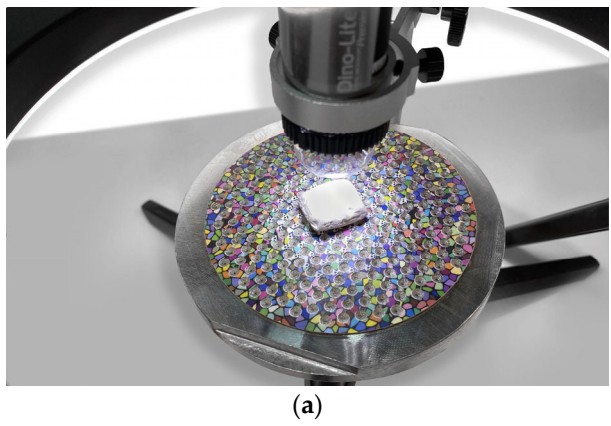

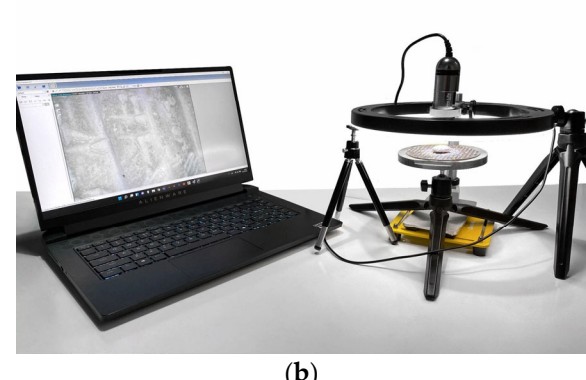

(**a**) (**b**)

**Figure 11.** The first acquisition configuration with the turntable calibrated plate: (**a**) three-dimensional calibrated plate (updated prototype) detail; (**b**) complete setup configuration.

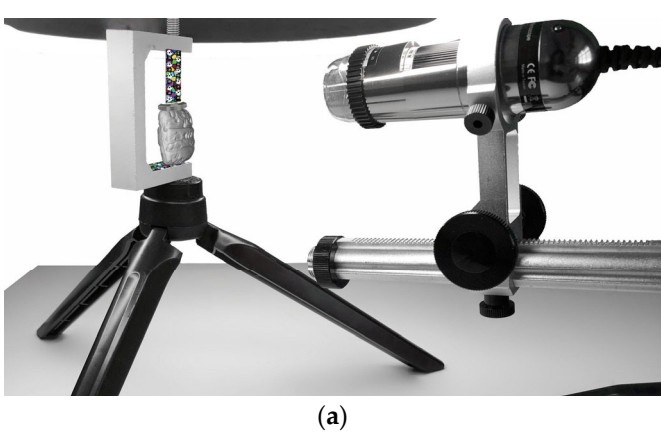

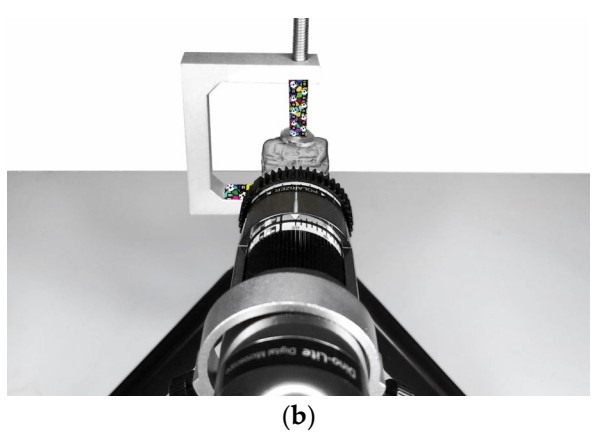

(**a**) (**b**)

**Figure 12.** The second acquisition configuration with the screw clamp: (**a**) the screw clamp coaxial to the rotating base; (**b**) detail of the positioning of the tablet in the clamp.

The Dino-Lite AM7013MZT was used for the tests and also in the next campaign to digitize the original cuneiform tablets. A 20× magnification was chosen to work with, at which the working distance is 48.7 mm, and the depth of field is 3.6 mm with this model.

The model has a built-in adjustable polarizer to reduce reflections on shiny objects. However, the lighting conditions were improved with the adoption of an LED illumination ring. The light does not directly hit the object due to the shape and diffusing material of the lens hood. Therefore, the diffused light conditions neutralize the shadow cones without changes in the intensity of the shadows, light, and colors. For the first configuration (vertical microscope axis), a calibrated circular plate was designed, obtained with an EOS M270, by additive manufacturing of steel powder and characterized by a pattern of 279 truncated conical holes, with a countersink angle of 60° at all times and a base diameter of 0.35 mm. Therefore, the coordinates of each hole in a local reference system are known, and the perforated plate can be used as a grid of constraint points (GCP) evenly distributed over the entire area to optimize camera orientation and scale the model.

The calibrated plate was placed on a rotating table to facilitate and speed up the acquisition phase.

Once the magnification rate (optical zoom) was chosen and the digital microscope was placed on the fixed bracket at a fixed angle, a series of images with sufficient overlap could be acquired by rotating the base. Thanks to the customized tripod, the inclination of the microscope related to the plate changes after each complete rotation.

In general, to overcome the problem of a shallow depth of field, one possibility would be to close the aperture of the optics as much as possible within the limits of the diffraction phenomena [42] and provide enough light to balance a correct exposure time. This option

is not available on digital microscopes, which have a fixed diaphragm and, as a result, a fixed sharp field. Therefore, the object is captured from different focusing planes by moving the microscope on the vertical micrometric rail to focus on different planes.

With this first updated configuration, two acquisition sets were required, one for the recto and the other for the verso of the tablet, with an average of 250 shots each and approximately 1 h of data acquisition work.

One of the problems that remained open for the digital survey of the cuneiform tablet (and thus for similar objects of small size and non-negligible thickness) was the impossibility of having a merged 3D model of the two faces using only the microscope dataset, even though inclined acquisitions were also carried out to frame the sides.

The second data acquisition setup was designed with a low-cost strategy in mind to address this specific gap. Indeed, the wedge-shaped tablet turned out to be a real challenge which, due to its size and complexity, allowed us to stress the acquisition system and bring out its weak points.

The second configuration involves a screw clamp commonly used in model making to position the object like a knife edge in relation to the microscope camera. The clamp was covered with an adhesive pattern with printed coded markers. As for the calibrated plate, the clamp was also placed on a rotating base, taking care to align the center of rotation with the mounting axis of the object. The other photogrammetric accessories remained virtually unchanged from the previous configuration.

This setup took roughly an hour to acquire 700 photographs, plus an additional set for the tiny parts concealed by the clamp (around 100 shots). By positioning the model in this second acquisition mode, we were able to lessen the issues brought on by the shallow depth of field by ensuring that the pattern of encoded markers was always coplanar to the plane of focus on the surface of the object.

In the first configuration, we had numerous problems connected with the impossibility of focusing the markers and the surface of the wedge-shaped tablet simultaneously. With this configuration, however, this problem was completely resolved.

Figure 13 displays the different performance abilities to simultaneously focus the markers and the object surface using the calibrated plate (a) or clamp (b).

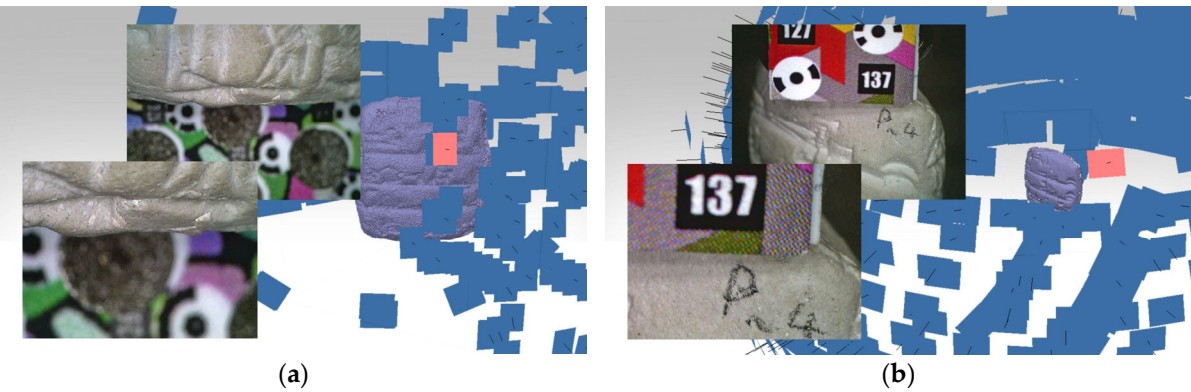

(**a**) (**b**)

**Figure 13.** The different performance abilities to simultaneously focus markers and object surface are compared using: (**a**) the calibrated plate or (**b**) the clamp.

The two datasets generated from the two acquisition settings were then processed using Structure from Motion (SfM) software (Agisoft Metashape Professional v.1.8.3), following the conventional photogrammetric workflow and using the same workstation (AMD Ryzen 9 5900 HX CPU and 32 GB RAM). Due to the short depth of field and uniform texture of the tablet, aligning the tilted photographs during the processing of the first set of acquisitions proved to be very challenging.

This issue was not present while using the second configuration with the clamp because it gave us a more straightforward, more regular (with a higher number of shots for the edges) and, consequently, more efficient capture geometry (Figure 14).

The GSDs for the two photogrammetric projects were 6.3 μm for the calibrated plate acquisition and 7.14 μm for the screw clamp acquisition, respectively. Table 2 contains the key details related to the photgrammetric image processing.

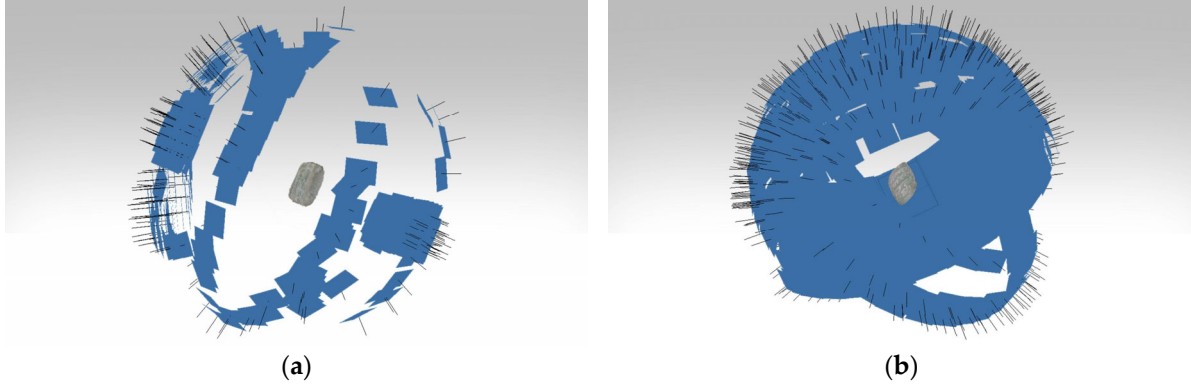

(**a**)          (**b**)

**Figure 14.** The capture geometry network corresponds to the use of: (**a**) the calibrated plate; (**b**) the screw clamp. The acquisition time is the same (1 h) even if the second configuration has more photos.

**Table 2.** Photogrammetric project properties.

| Acquisition Network Geometry | Calibrated Plate Set | Screw Clamp Set |
| --- | --- | --- |
| Aligned Cameras | 506/518 | 718/718 |
| Sparse Cloud | $4.3 \times 10^5$ points | $4.1 \times 10^5$ points |
| Sparse Cloud Filtered | $3.8 \times 10^5$ points | $2.1 \times 10^5$ points |
| Dense Cloud | $10.4 \times 10^5$ points | $8.3 \times 10^5$ points |
| GSD | 6.3 μm/px | 7.14 μm/px |
| RMS Error | 0.87 mm | 0.027 mm |
| Processing Time | 4 h | 2 h |

### 3.3. Benchmark and Evaluation Comparisons

The tablet 724 replica was also digitally surveyed with a structured light scanner (Scan in a Box @2015 Open Technologies SRL, Brescia, Italy) in order to have a digital reference model against which to check the overall geometric dimensions of the photogrammetric reconstructions generated from the previous tests. The final mesh model, which has an average resolution of 0.08 mm and is made up of around 400,000 polygons, has fewer details than the corresponding photogrammetric models but can still be utilized to ensure a reference check of their overall dimensions (Figure 15).

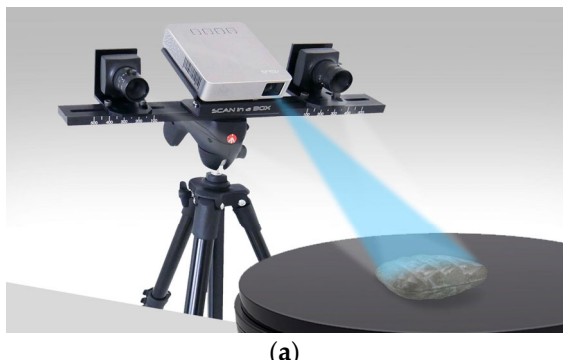 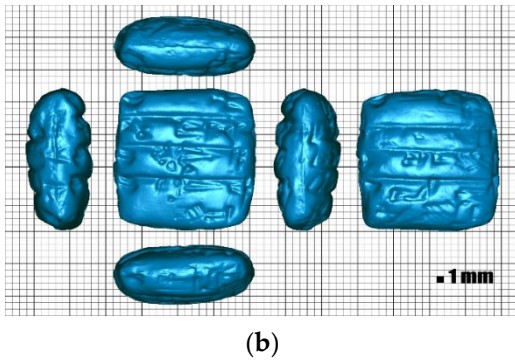

(**a**)          (**b**)

**Figure 15.** The benchmark survey: (**a**) the Scan in a Box system; (**b**) the polygonal model of the cuneiform tablet replica (1 mm grid background).

The scanner mesh model was used as a reference to align the photogrammetric clouds in the same coordinate system and to provide a comparison of the results obtained.

In this analysis, the photogrammetric models obtained with the two configurations described in Figures 11 and 12 were evaluated: the first based on the use of a calibrated plate on which to place the object, the second on the use of a clamp in which to fix the object, covered by a calibration pattern.

The results of the first basic setup acquisition, shown in Figure 10 [11], were excluded, and only the optimized configurations were considered in order to choose the better one for this case study. Cloud-to-mesh registration (CloudCompare v. 2.12 alpha, open source software) was performed in two stages: a manual registration using homologous points and a global registration using automatic alignment algorithms (iterative closest point).

Figure 16 displays the false-color deviations between the photogrammetric clouds and the laser mesh reference. It should be observed that for both photogrammetric models, the mean and standard deviations from the reference do not exceed 0.3 mm.

Moreover, it can be seen from the image that many deviations are punctual and are mainly localized in some deeper incisions. Therefore, there are probably no differences in the overall dimensions of the tablet or the deformations in the morphology. Most of these deviations seem related to the higher resolution obtained from the photogrammetric survey.

The scanner mesh is smoother and has a lower resolution: 0.08 mm against 0.007 mm GSD. These considerations allow us to conclude that the microscope acquisitions have been correctly processed within the photogrammetric workflow and that the digital models are satisfactory both from the metric point of view and for the subsequent interpretation of the data.

Concentrating instead on the comparison between the model generated by the two network geometries, Figure 17 illustrates that for the screw clamp setup, 94% of the points are included in a difference of ±0.1 mm, while about 40% for the calibrated plate configuration are included in a difference of ±0.1 mm.

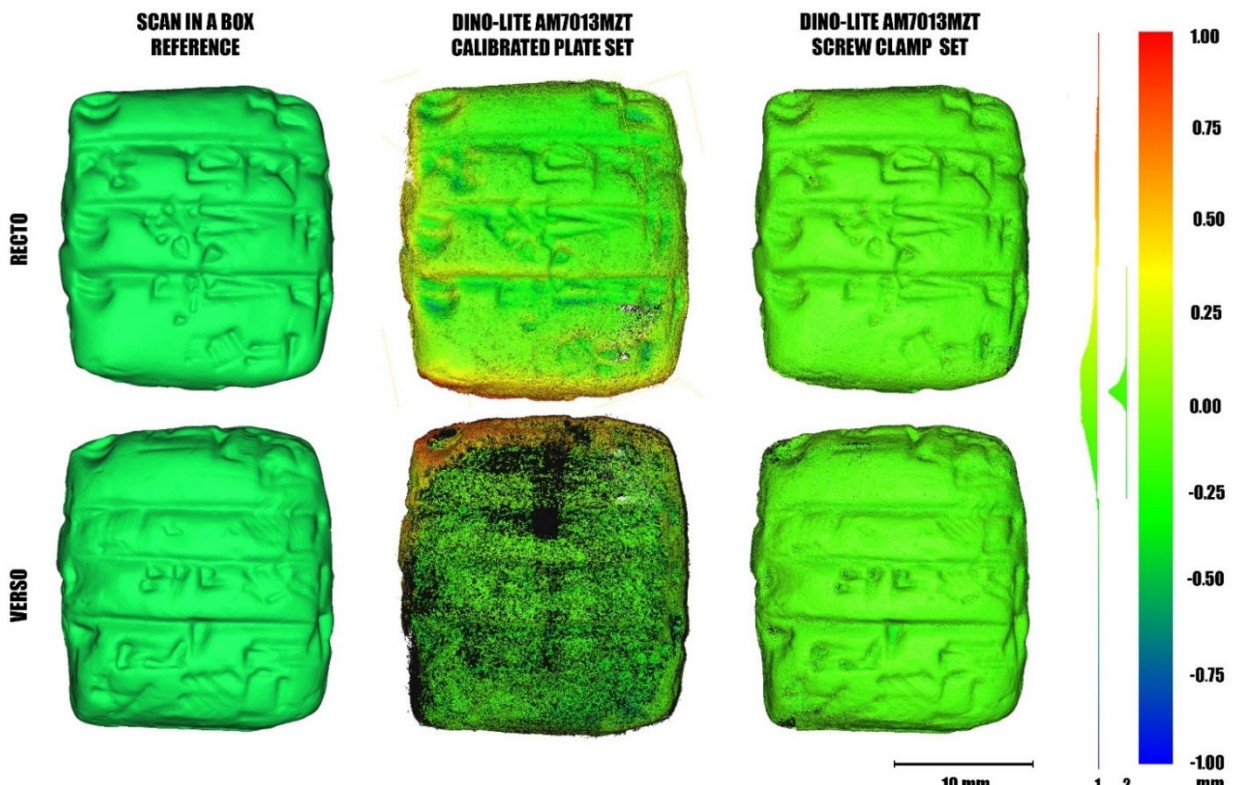

**Figure 16.** False-color deviations from the Scan in a Box reference model of the photogrammetric dense clouds obtained with the calibrated plate and screw clamp as support (recto and verso side).

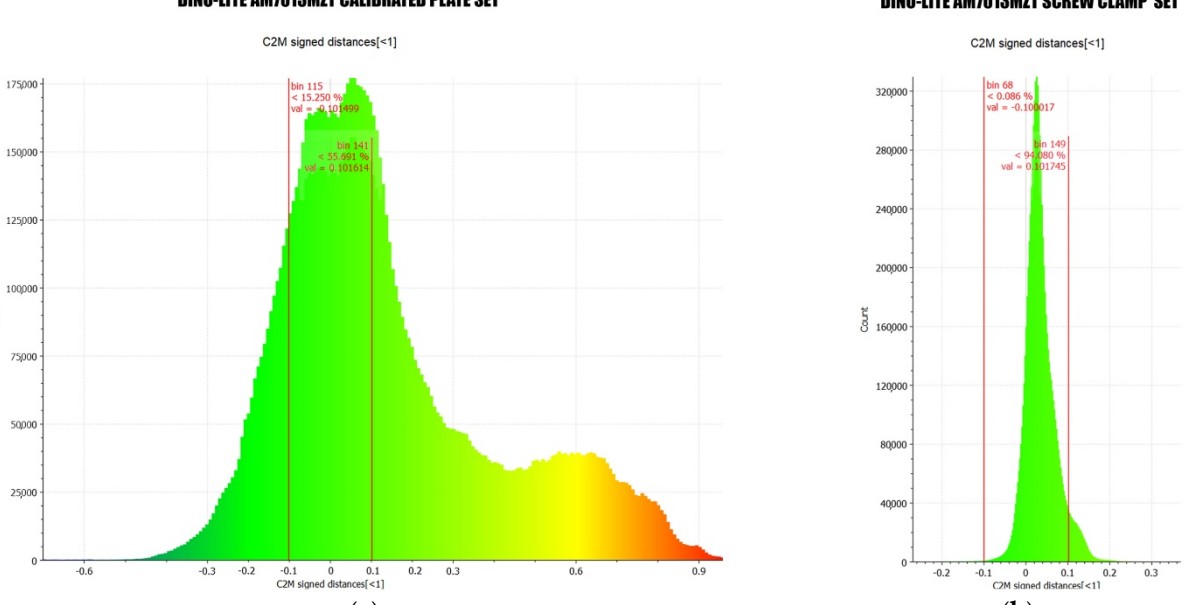

(**a**)            (**b**)

**Figure 17.** Deviations of the photogrammetric clouds from the reference model: (**a**) in the calibrated plate setup, 40% of the points are within ±0.1 mm; (**b**) in the screw clamp setup, 94% of the points are within ±0.1 mm.

In depth, the results produced using the calibrated plate show more considerable deviations. This is because of problems with the photo alignment, which have made the cloud noise worse (in the lower right corner of the recto and on the whole face of the verso) and caused difficulty in the merging of the two sides (Figure 18).

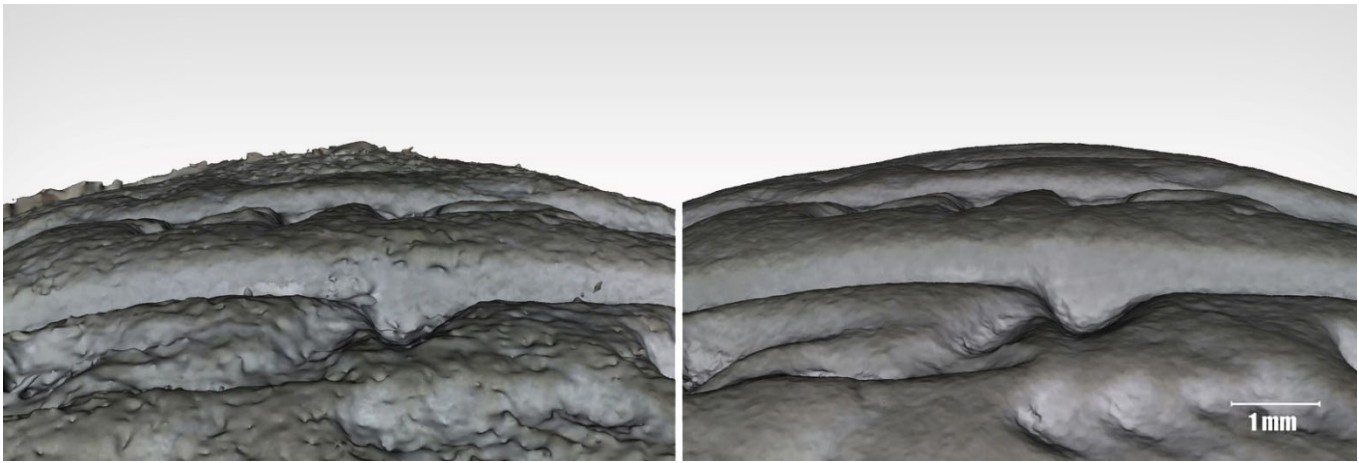

**Figure 18.** Comparison between the surface of the models; the reconstruction related to the first setup acquisition with the calibrated plate presents more noise.

In conclusion, the calibrated plate remains a valuable tool for the predominantly flat objects and/or those combined with reflex cameras and macro lenses, allowing for greater depth of field and a wider angle of view.

However, as is known, it is not possible to define a photogrammetric survey configuration appropriate to all scales, sizes, and morphologies [43] (p. 443).

The availability of different supports and accessories (tripods, light setup, calibrators) allows us to quickly change the acquisition geometry or even the acquisition tool (camera, cell phone, and microscope).

Therefore, the system has been built to be flexible, such that alternative network geometries are conceivable using the same equipment for data collecting (in this case, the

microscope) by selecting the most appropriate support based on the shape and features of the object.

## 4. USB Microscope Optical Calibration

### 4.1. Premise

The 'so-called' USB microscopes constitute an optical inspection system designed to obtain as output the image on a monitor, possibly by means of dedicated software [44]. As is shown, the magnifying capacity and the ability to be tilted, given the shape and small size, enable images with appropriate intersection angles to the object to be obtained for the generation of three-dimensional models [40] according to the known SfM procedures.

One of the main problems in SfM concerns the identification of two essential items: (i) the position and orientation of the camera in space (the external parameters of the camera) and (ii) how the camera maps the perspective projection points of space onto its image plane (the internal parameters). The determination of these parameters is significant for the accuracy of the 3D reconstructions, as well as for reducing the number of unknowns in the collinearity equations and thus speeding up the bundle-block adjustment procedure.

While the software available on the market today has made enormous strides and allows for fairly accurate three-dimensional models, even for uncalibrated cameras within a self-calibrating bundle adjustment [45], the effectiveness of the reconstruction obtained requires validation.

The question then arises as to whether it is possible to define the optical parameters of the Dino-Lite microscope (in detail with reference to the AM7013MZT model used for this research activity), operating in the lack of EXIF data and thus just starting from the known values stated by the manufacturer. In detail, the focus is on the identification of a geometric model compatible with the USB microscope used and the estimation and verification of the main distance. It should be noted that the only a priori known data relating to the optical system are the sensor size in pixels (5.0 MP, 2592 × 1944 px) and, depending on the magnification rate (M)[4], the field of view (FOV) and the working distance (WD)[5] [46]; see Table 3.

**Table 3.** Working distance (WD) and field of view dimensions (FOV) in the x and y directions as a function of magnification rate (M) for AM7013MZT Dino-Lite microscope expressed in millimeters.

| M | WD | $X_{FOV}$ | $Y_{FOV}$ |
|---|----|-----------|-----------|
| 20 | 48.7 | 19.8 | 14.9 |
| 30 | 21.7 | 13.2 | 9.9 |
| 40 | 9.0 | 9.9 | 7.4 |
| 50 | 1.9 | 7.9 | 5.9 |
| 60 | −2.3 | 6.6 | 5.0 |
| 220 | −0.1 | 1.8 | 1.4 |
| 230 | 1.0 | 1.7 | 1.3 |
| 240 | 2.1 | 1.7 | 1.2 |

To develop a method for estimating the unknown parameters of an optical system for photogrammetric purposes, a model that acceptably synthesizes the geometry of the system must be assumed. The type of model used performs according to the same laws as the pinhole camera model (perspective projection) and follows the Gaussian lens (thin lens) law.

### 4.2. Optical Model

Following the geometrical optics, the image formation using a lens provides a relationship between the object in real space and its image on the sensor plane.

In general, the distance of the object from the lens called $H$, and the distance of the sensor from the lens, called $f$[6], are in a conjugate proportion (similar triangles). Observing this general law, it is then possible to admit that the field framed at a given magnification, i.e., for a given position of the optical unit[7], is related to the image framed totally by the sensor (Figure 19).

In other words, if a graduated bar is photographed at a given magnification, the framed—and thus visible—field of the image corresponds to a specific, quantifiable length, called field of view, along the horizontal (XFOV) and the vertical direction (YFOV) of the sensor.

This is a very important piece of information, easily verifiable experimentally, allowing the field framed at each magnification to be related to the size of the sensor, as follows:

$$|X_{FOV}/X_S| = |H/f| \tag{1}$$

$$|Y_{FOV}/Y_S| = |H/f| \tag{2}$$

Unfortunately, Equations (1) and (2) require the dimensions in millimeters of the sensor to be known. For now, the discussion on this is postponed to the next paragraph.

The first thing one can do, however, is to calculate the so-called GSD, "Ground Sample Distance" from the ratio $|X_{FOV}/X_S|$, expressed in millimeters divided by pixels, choosing a magnification from the microscope dial, setting up at the correct working distance, and photographing a graduated bar, as previously indicated.

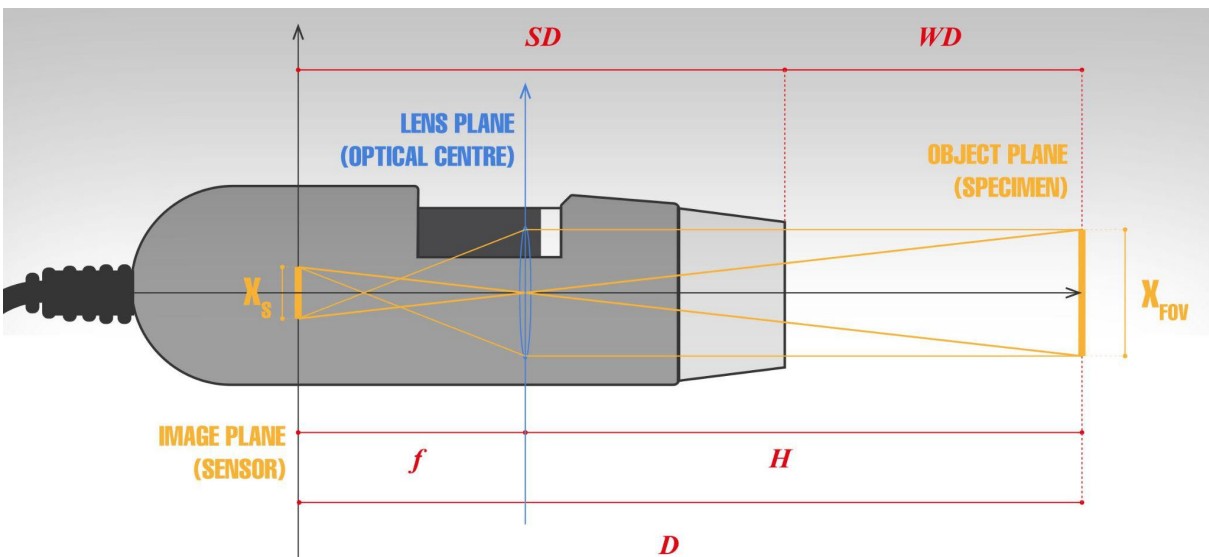

**Figure 19.** Image formation model for a fixed magnification factor in USB digital microscopes. The following quantities are highlighted: $f$—the distance of the sensor from the lens, also called main distance; $H$—the distance of the object from the lens; $WD$—the distance of the microscope nozzle from the specimen to focus correctly; $SD$—the position of the sensor with respect to the microscope nozzle.

For instance, the values for 20× magnification are considered as follows:

$$GSD = X_{FOV}/X_S = 19.8/2592 = 0.00764 \text{ mm/px} \tag{3}$$

At this point, proceeding experimentally in the Agisoft Metashape (pro. v.1.8.3) software environment, it is possible to derive the missing data of Equation (1,2) via the "calibrate lens" function, using a printed calibration target accepted by the software, consisting of a square chessboard with a 1 mm grid step.

Ensuring that the magnification, i.e., the framed field of view, is 19.8 mm in the x direction, the chessboard is then framed and photographed from different angles; the aim is to acquire a number of images greater than 10. The lens calibration procedure supports

the full camera calibration matrix estimation, including the non-linear distortion coefficients [47]. The estimated parameters pertain to Brown's model: $f$—main distance measured in pixels; $c_x$, $c_y$—principal point coordinates; $k_1$, $k_2$, $k_3$—radial distortion coefficients; and $p_1$, $p_2$—tangential distortion coefficients.

Following the process of the lens calibration steps, image orientation, and scaling and referencing in a local coordinate system according to the grid, the calibration parameters of the camera were extrapolated.

Both the known a priori parameters and those obtained from the calibration process for the 20x magnification are then summarized below (Table 4).

**Table 4.** Declared data and computed data after calibration (besides distortions and coordinates of the principal point) for 20x magnification.

| Declared Parameters | | | | | Calibration Output Parameters | | |
|---|---|---|---|---|---|---|---|
| **WD** | **$X_S$** | **$Y_S$** | **$X_{FOV}$** | **$Y_{FOV}$** | **GSD** | **$f$** | **$H$** |
| 48.7 mm | 2592 px | 1944 px | 19.8 mm | 14.9 mm | 0.0076 mm/px | 13413 px | 102.46 mm |

Note that at this stage, $f$ is relevant regarding internal orientation and H and GSD are relevant regarding data derived from external orientation. It is clear that the lens calibration output parameters may fluctuate, albeit slightly, as experimentally obtained data from the procedure described above. The additional difficulty of manually setting the magnification to perfectly match the 20× value indicated on the control ring can also influence variations in the results. From the table, it can be seen that the GSD value calculated by the software is coincident with that calculated a priori from both the declared values in Equation (3).

Having the main distance calculated by the software in pixels—the size of the main distance in millimeters remains unknown because the pixel pitch value is unknown—the value of $H$ for the assumed camera model can be verified from Equation (1) as follows:

$$H = f \times (X_{FOV}/X_S) = f \times GSD = 102.48 \text{ mm} \tag{4}$$

It can be seen that the two values of $H$, the first in Table 2 and the second obtained from Equation (4), are very close.

So, it is possible to accept $H$ = 102.5 mm as an exact value.

*4.3. Sensor Size*

At this point, the issue of sensor size needs to be clarified. In the absence of information on the size in millimeters of the entire sensor, and thus the pixel pitch, and on its geometric location in the model, it is not possible to identify it with certainty.

Even if one wanted to proceed iteratively, any sensor among those on the market and compatible with the geometry of the microscope would have an acceptable solution: this is because, in the absence of a metric reference of the sensor, it is understood that 2592 px along the x sensor direction can be accommodated by any sensor that physically fits into the microscope: [48] (Table 5).

**Table 5.** Different sensor sizes potentially suitable for the model assumed and the derived *f* in the pixel for 20× magnification. For each sensor, the value of *f* in mm was derived by multiplying the *f* shown in Table 2 per the pixel pitch. It is not a surprise that *f* in millimeters changes since (1) shows the direct proportionality between $X_S$ and *f*.

| Sensor Type | $X_S$ | | $Y_S$ | | Pixel Pitch | $F$ | |
|---|---|---|---|---|---|---|---|
| 1/9″ | 2592 px | 1.60 mm | 1944 px | 1.20 mm | 0.0006 mm/px | 13,413 px | 8.28 mm |
| 1/6″ | 2592 px | 2.40 mm | 1944 px | 1.80 mm | 0.0009 mm/px | 13,413 px | 12.42 mm |
| 1/4″ | 2592 px | 3.60 mm | 1944 px | 2.70 mm | 0.0014 mm/px | 13,413 px | 18.63 mm |
| 1/3″ | 2592 px | 2.80 mm | 1944 px | 3.60 mm | 0.0019 mm/px | 13,413 px | 24.84 mm |
| 1/2.5″ | 2592 px | 5.76 mm | 1944 px | 4.32 mm | 0.0022 mm/px | 13,413 px | 29.81 mm |

Basically, a scaling factor (the pixel pitch) determines the focal length in mm and the subsequently calculated quantities.

Consequently, it was necessary to contact the manufacturer to know the sensor size: the sensor of the AM7013MZT measures 3.67 × 2.74 mm, resulting in a pixel pitch of 0.0014 mm, values compatible with a 1/4″ sensor.

Let a further quantity be introduced: this is a constant quantity called SD, which defines the position of the sensor with respect to a known point on the model, a point which can be assumed for convenience to coincide with the tip of the front cap. SD is defined as:

$$SD + WD = f + H = D \tag{5}$$

It is therefore trivial to calculate the value for the given sensor when deriving that the position of the sensor is approximately 72 mm anterior to the end of the microscope, a value further confirmed by the manufacturer (Table 6):

**Table 6.** Values of characteristic quantities for the sensor used in the USB microscope from calibration test.

| Sensor Type | WD | *f* | *H* | D | SD |
|---|---|---|---|---|---|
| 1/4″ | 48.70 mm | 18.6 mm | 102.5 mm | 121.1 mm | 72.4 mm |

*4.4. Result Validation*

With the aim of performing a final cross-check with the initial calibration, the parameter estimation is then repeated, assuming that the sensor is known from the beginning. Indeed, it should be recalled that the initial lens calibration was performed without EXIF data, i.e., the software only estimated the quantities from the features recognized on the calibration grid and the grid step size. Therefore, the sensor input data can be used to verify the validity of the estimation made by the lens calibration tool. Then, the ratio of the framed field to the corresponding sensor size in millimeters for the two magnifications considered previously is computed as follows:

$$X_{FOV}/X_S = 19.8/3.67 = 5.4 \tag{6}$$

By noting that the WD for each magnification is known and experimentally verifiable and that SD is declared by the manufacturer to be equal to 72 mm, the sum of SD and WD is known; so, Equation (5) can be applied. Then, the following systems can be formulated and solved for the unknowns *f* and *H*:

$$f + H = 121.1 \tag{7}$$

$$H/f = 5.4 \tag{8}$$

Then, the values obtained from this verification are compared with those of the calibration (Table 7). The differences related to the software calibration can be considered

negligible, and the result obtained by the software is considered as acceptable, as is the model used.

**Table 7.** Values of characteristic quantities for 1/4″ sensor at 20× magnification in the Dino-Lite AM7013MZT USB microscope.

| Values | $f$ | | $H$ | GSD |
|---|---|---|---|---|
| Calibrated | 13413 px | 18.6 mm | 102.5 mm | 0.00764 mm/px |
| Calculated | 13481 px | 18.9 mm | 101.8 mm | 0.00764 mm px |

## 5. Massive Cuneiform Tablets Digitization

The tests described in the preceding paragraphs (3 and 4) highlighted the problems faced when surveying small objects by USB microscopes, which have driven the evolution of the solutions adopted to streamline the procedures, paying attention to optical calibration. It then emerged that the network geometry obtained with the screw clamp setup was much more effective than the nadiral one.

Indeed, despite the tablet shape and morphology, it ensures sufficient overlap between the pairs of stereoscopic images, improving the camera orientation, allowing a complete and continuous reconstruction of the surface, and facilitating and speeding up the acquisition.

Therefore, it was possible to carry out expeditious digitization of the original tablets at the University of Ghent by the screw clamp setup, including the microscope Dino-Lite AM7013MZT, a LED illumination ring, and a turntable (Figure 20).

The resulting less noisy models solved two critical issues: (i) there were no severe problems focusing the tablet surface and the markers (fixed to the stand) at the same time, despite the small depth of field and (ii) the front and back of the tablet could both be simultaneously captured. Although it was necessary to turn the piece upside down and partially repeat the capture process for the occluded part, the alignment of this second set of photos was not problematic, guaranteeing legible and complete final models.

In a first survey campaign, four cuneiform tablets were digitized in two full days of work, with an average of two complete tablets per day (Table 8). The images of each dataset were then imported into the Agisoft Metashape (pro. v.1.8.3) software for SfM processing.

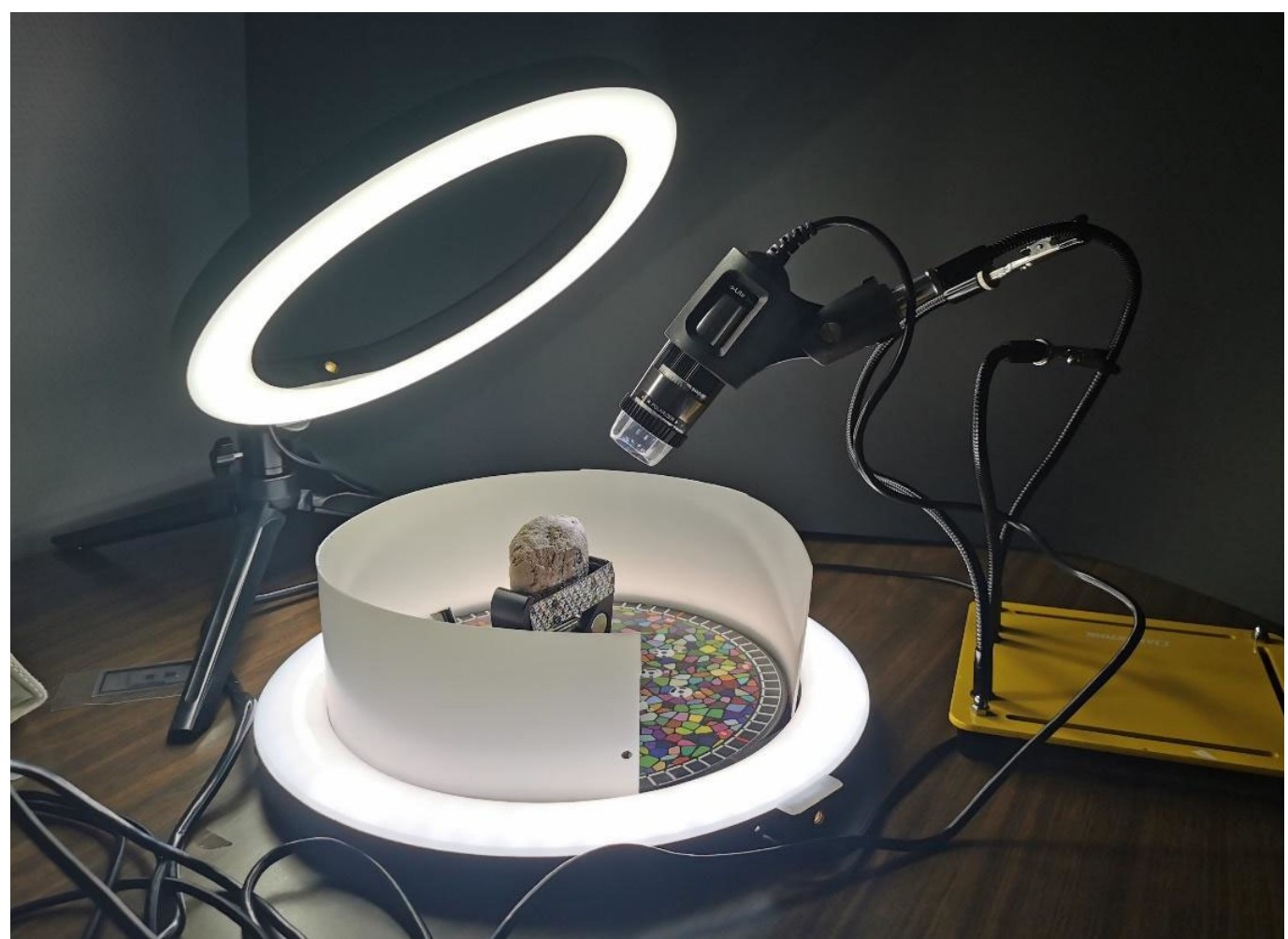

**Figure 20.** Acquisition system based on the turntable and clamp with ring illumination to support the microscope-integrated LED illumination. Magnification related to working distance: 20×.

**Table 8.** Cuneiform tablet digitized at Ghent University with the USB microscope Dino-Lite AM7013MZT.

| Tablet LW21.CUN. | 160 | 159 | 133 | 126 |
|---|---|---|---|---|
| Captures number | 959 | 930 | 949 | 1343 |
| GSD | 0.00752 mm/px | 0.00781 mm/px | 0.00766 mm/px | 0.00766 mm/px |
| Recto | | | | |
| Verso | | | | |

The internal and external orientation parameters were calculated during the image alignment step with the bundle block adjustment process based on the collinearity equations [49]. The computer vision SfM algorithm converges faster to a solution if it starts from input data about the optical calibration close to the real ones.

Unfortunately, this information could not be automatically deducted from the EXIF data as they were absent. Therefore, the initial approximation of the interior orientation parameters was specified before starting the process according to the calibration results (see paragraph 4).

Subsequently, it was possible to identify the encoded targets in each capture automatically. To perform this function, it was necessary to design a pattern of software-generated targets to be introduced into the scene based on the object size [50].

However, the need to include metric references or calibration tools in the captured scene, using the appropriate patterns in the function of the object dimension, still represents a critical issue: indeed, it is not trivial to estimate the accuracy of the print and ensure the flatness of the reference surface, as well as to position the targets themselves simultaneously close to the object and within the sharp area.

A first orientation optimization is then based on the markers automatically generated by the recognition of these encoded targets. This operation simultaneously refines the external and internal camera orientation parameters and the coordinates of the triangulated constraint points. At this stage, the local coordinates for each marker are not specified, although known by design, in order not to stiffen the model geometry with the introduction of errors related to the marker printing accuracy.

A second optimization is performed after filtering the sparse cloud points (tie points) based on specific criteria. The parameters considered are: (i) reconstruction uncertainty; (ii) reprojection error; and (iii) projection accuracy.

The typical consequence of the points generated from nearby photographs with a thin baseline is a great amount of reconstruction uncertainty. These points may create a noticeable noise on the surface with variable density. Even though removing such points does not affect optimization accuracy, it may be advantageous to improve the clearness and correctness of the reconstruction geometry.

High reprojection error typically indicates poor localization accuracy of the corresponding point projections during the point matching step. False matches are also common. Eliminating these spots can increase the accuracy of the next optimization stage. The projection accuracy criterion enables the removal of points whose localization of projections was substantially poorer due to their larger size.

Therefore, the point cloud filtering tools allowed us to distinguish and remove the points that substantially impacted the reconstruction quality.

After the filtering-based optimization, the coordinates of the markers were entered for the sole purpose of scaling the model (absolute orientation). The accuracy given by the RMSE (root mean square error)[8] of the measured image coordinates was always less than 0.5 pixels in all eight chunks of the tested cases, with the most significant total displacement error between 1 and 2 tenths of a mm.

Next, following the canonical steps of photogrammetric reconstruction, the dense cloud, mesh, and model texturing were generated. Two chunks were created for each tablet because it was necessary to integrate the first complete round of acquisitions with another set of photos of the area occluded by the support and the pattern.

The two groups were then aligned using homologous natural features (marker-based) to obtain a complete reconstruction of each tablet. It should be emphasized that the joining chunks may sometimes be imperfect, invalidating the final result, with merging errors that must subsequently be post-produced (Figure 21).

For this reason, the study of the overlapping areas should be carefully planned upstream of the acquisition. In this regard, the safe and complete overlap work is still laborious due to minimal movement difficulty, which is controlled without the aid of micrometric slides; these should therefore be included in the system soon.

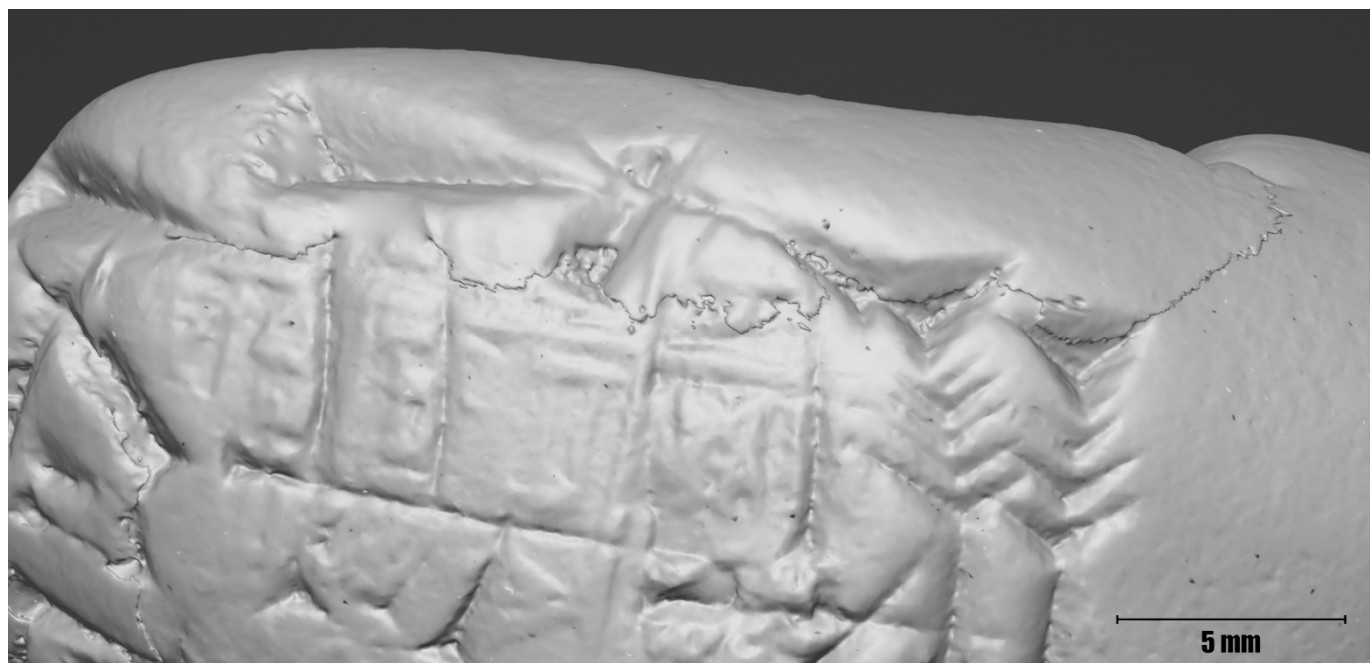

**Figure 21.** Detail of the mesh obtained by joining two partial models; evident is the stitching of the two pieces, which can be smoothened in post-processing.

Additionally, the shallow depth of field (due to the magnification increasing) implies that only a tiny portion of the artifact is in focus in the photo; so, only a small part of the image appears sharp enough to be used for 3D reconstruction [51].

The photo sharpness is one of the parameters determining the output point cloud reliability, accuracy, and quality [52]. So, computational photography techniques to extend the sharp field, such as the well-known focus stacking, allow less noisy results with more incisive surface detail [53]. Unfortunately, this is undoubtedly time-consuming and does not lead to time optimization. In all four cases examined, however, it was possible to achieve the completeness of the model without severe criticalities, thus ensuring the optimal three-dimensional legibility of the tablets (Figures 22–24).

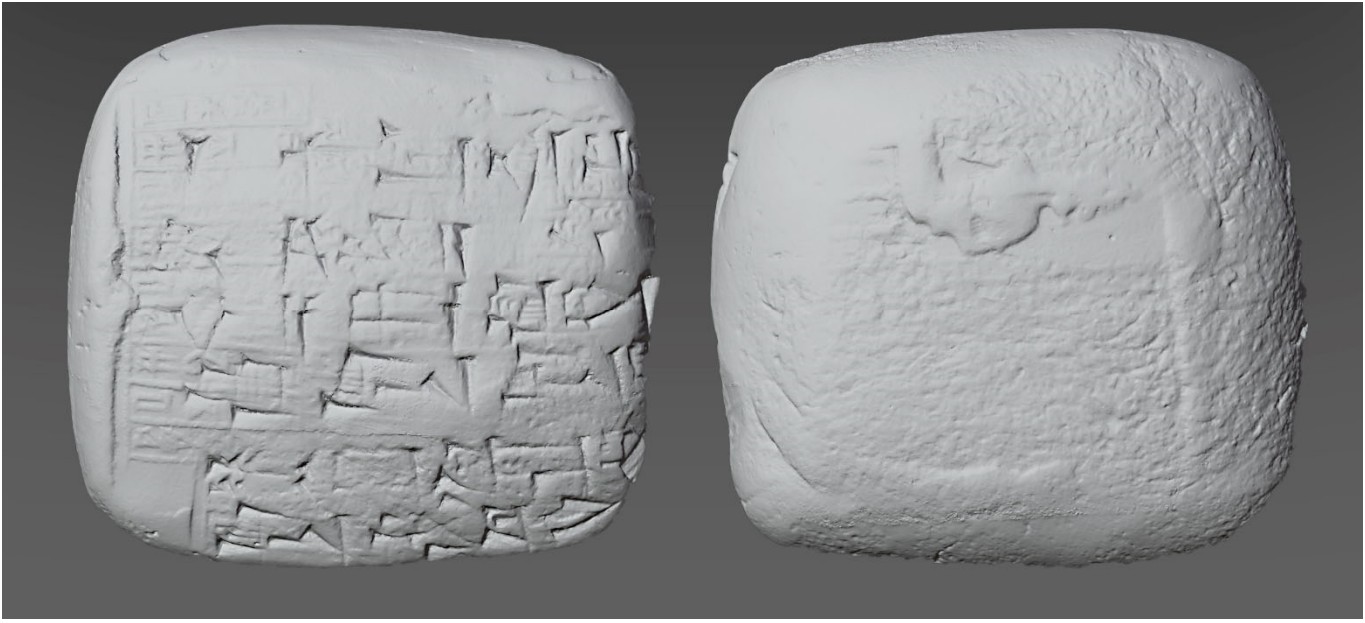

**Figure 22.** Complete mesh model of the Tablet LW21.CUN.133, obtained by photogrammetric survey with the Dino-Lite AM7013MZT microscope.

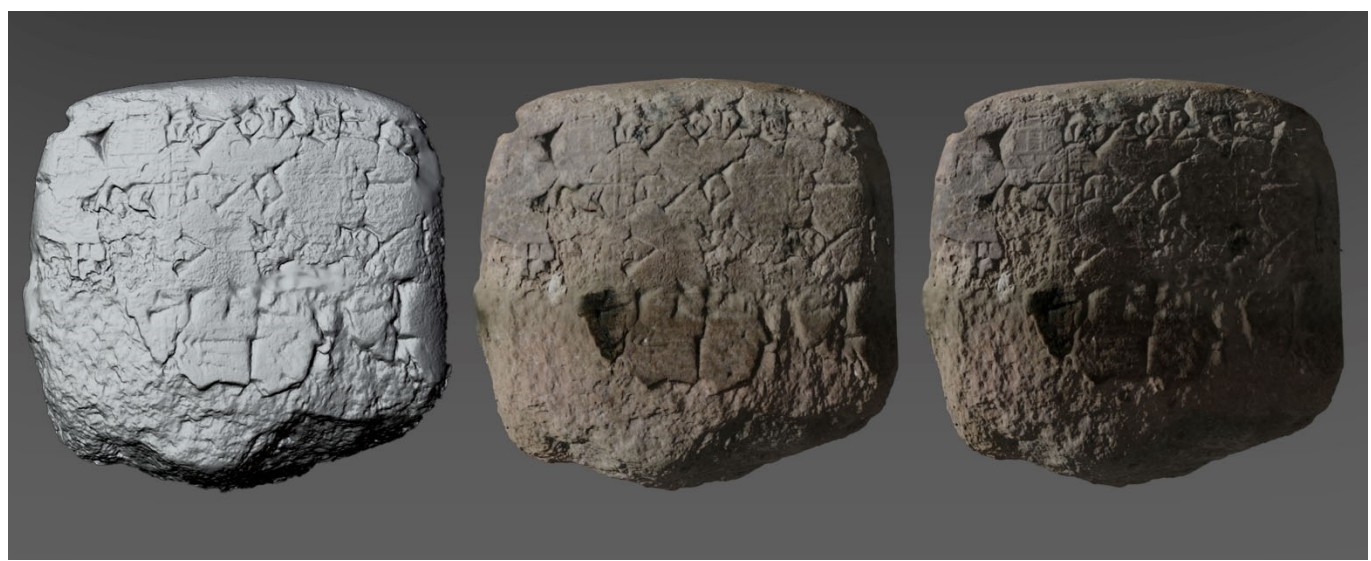

**Figure 23.** Complete mesh model of Tablet LW21.CUN.126, obtained by photogrammetric survey with the Dino-Lite AM7013MZT microscope, under different light conditions: diffuse light on mesh only; semi-grazing light on mesh and texture; highly grazing light on mesh and texture.

**Figure 24.** Complete mesh model of Tablet LW21.CUN.160 (**left**) and Tablet LW21.CUN.159 (**right**), obtained by photogrammetric survey with the Dino-Lite AM7013MZT microscope.

As already performed for the preliminary tests (Section 3), comparing the outputs of the different instruments used during the massive acquisition phase was helpful for the following reflections. Photogrammetry emerges as an excellent alternative to structured light scanning and can be conducted with valid results, not only by using consumer cameras (suitably equipped to achieve high magnifications) but also with USB microscopes of a few hundred euros.

However, the comparison of the digital models of the same tablet (Figure 25) shows that, despite the accuracy achieved by the different instruments in metric terms for the reconstruction, the size of the sensor and the quality of the optics used still have a clear impact on the rendering of the details (the camera model has sharper details than the others). Therefore, high magnifications are not synonymous with higher quality [54], especially if the depth of field, an essential photogrammetric requirement, is sacrificed.

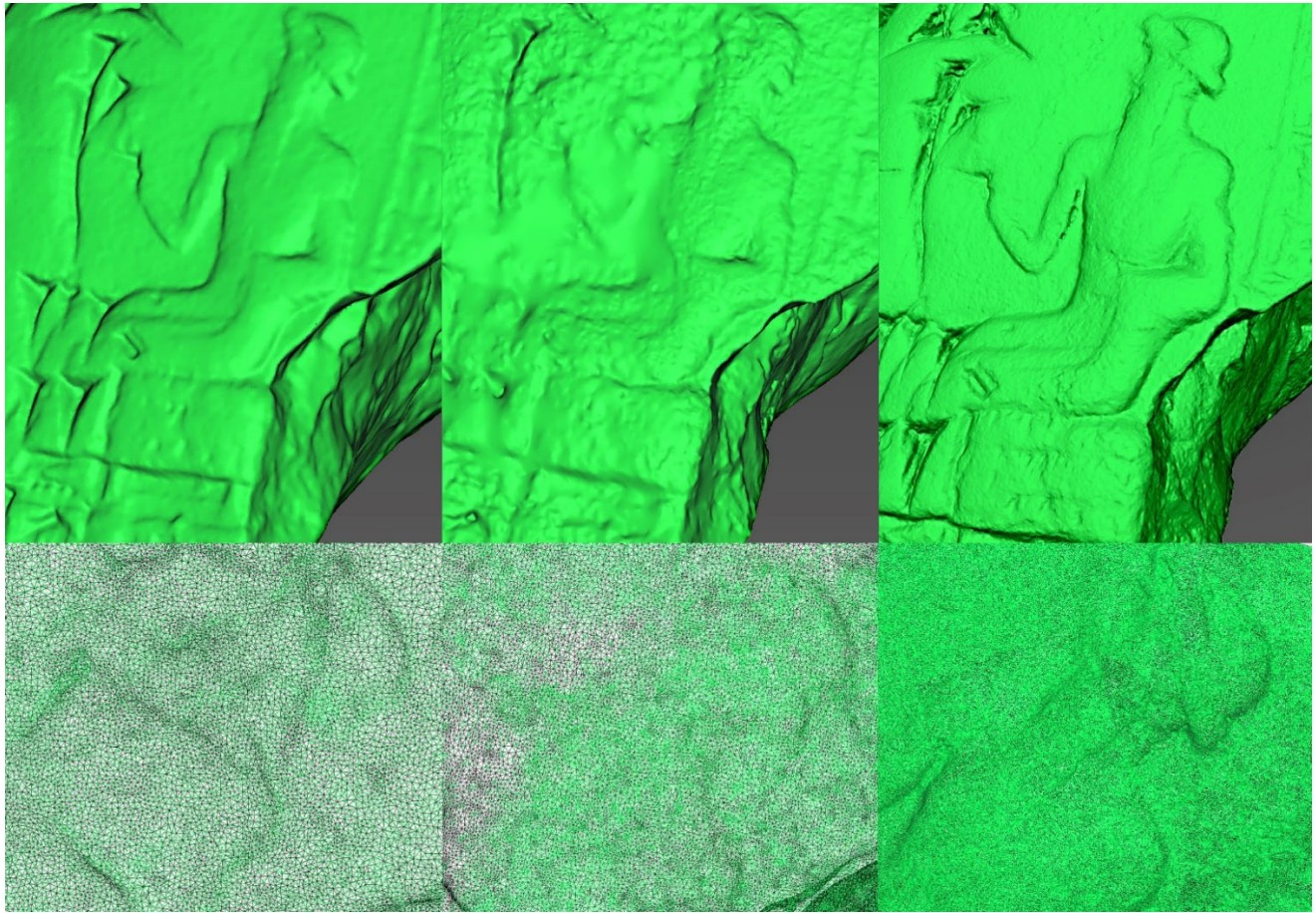

**Figure 25.** Detail and wireframe zoom of the Tablet LW21.CUN.159 verso, obtained by surveying with Scan in a Box 3D Scanner (**left**); Dino-Lite AM7013MZT microscope (**center**); DSLR Camera Nikon D800E combined with AF-S Micro NIKKOR 60 mm f/2.8 G ED (**right**).

## 6. Conclusions

The initial examination of the various conventional and innovative techniques for the documentation of the wedges on the cuneiform tablets has allowed us to focus on the specific needs of this type of study. In general, the digital micro-survey and the related polygonal model can be considered an optimal solutions for the reading and interpretation of the text, as well as for the possible sharing with other researchers and research studies. The solution proposed for the photogrammetric detail survey based on a customized configuration for a digital microscope was very effective.

The various experimental survey campaigns made it possible to define the "screw clamp" as the most suitable and fastest digital microscope acquisition setup in the function of the object shape and the required detail (see paragraphs 3 and 5). Figures 23 and 24 show the obtaining of a complete textured reconstruction. The resulting accuracy and the geometrical model correctness were validated using a reference benchmark (Section 3). In conclusion, the presented micro-photogrammetric configuration allows for the meeting of various needs: a low cost of equipment, high resolution (7–8 microns), and the color of the mesh as additional information to the geometry.

**Author Contributions:** All authors conceived and designed the experimental survey, investigation, methodology, and conceptualization; data acquisition and software, S.A.; validation F.F.; formal analysis S.A. and F.F.; resources and case study M.S.; data curation and visualization S.A.; supervision F.F.; project administration M.S.; heritage digitalization methods and benchmark F.F.; cuneiform tablet documentation M.S.; macro-photogrammetry configuration S.A. All authors reviewed and edited the entire paper. All authors have read and agreed to the published version of the manuscript.

**Funding:** This research was funded by Fonds Wetenschappelijk Onderzoek-Vlaanderen (FWO), grant number 11B8821N.

**Institutional Review Board Statement:** Not applicable.

**Informed Consent Statement:** Not applicable.

**Data Availability Statement:** Not applicable.

**Acknowledgments:** We would like to thank, sincerely, IDCP Digital Innovation—in the persons of Jan Boers, Danielle van Duijvendijk, and Ivo Manders—and, of course, Dino-Lite Digital Microscope for the support, availability, and equipment offered for the research. Special thanks to Diego Ronchi (ISPC, Consiglio Nazionale delle Ricerche); Katrien De Graef (Ghent University), Catherine Mittermayer (University of Geneva), and Marinella Levi (+LAB, Politecnico di Milano) for sharing dataset, materials, and research; and Fabrizia Caiazzo, Vittorio Alfieri, and Paolo Argenio for the prototype of the circular steel calibrator. Thanks to the 3D SurveyGroup (ABC Department, Politecnico di Milano) for sharing experience, know-how, and equipment.

**Conflicts of Interest:** The authors declare no conflict of interest. The funders had no role in the design of the study; in the collection, analyses, or interpretation of data; or in the writing of the manuscript.

## Note

1. The most common and most expensive micro-survey solution is the structured light scanner, which can cost upwards of EUR 30.000 at the highest standards (without taking into account the costs of licenses for data management software). In general, it allows the reaching of accuracies of 0.01 mm and resolutions down to 0.03 mm [https://www.artec3d.com/it/portable-3d-scanners/artec-micro; https://www.einscan.com/handheld-3d-scanner/einscan-pro-hd/; https://www.creaform3d.com/en/portable-3d-scanner-handyscan-3d; https://www.hexagonmi.com/products/structured-light-scanners; https://lmi3d.com/family/line-profile-sensors/]. However, today, various low-cost 3D scanners (from EUR 500) are also available; these are naturally designed for non-professional applications, with correspondingly much poorer accuracy and resolution and for mostly amateur use [https://www.xyzprinting.com/en/product/3d-scanner-2-0; https://it.shop.revopoint3d.com/; https://www.creality3dofficial.eu/collections/scanner-&-engraver-eu; https://scandimension.com/products/sol-3d-scanner; https://matterandform.net/store].

2. As a rough estimate, the price of a Dino-Lite microscope begins at a few hundred euros and varies based on the technical specs (resolution, magnification, LEDs, polarize, etc.). A mid-range one costs approximately EUR 500. In contrast, the price of a set for micro-photogrammetry varies widely based on the technical data (sensor size, resolution, lens quality, accessories for macro photography, etc.). However, the price of a high-level reflex camera+macro lens typically exceeds EUR 3000 (the cost estimation is refereed in Europe).

3. The digital survey took place in Geneva (Switzerland) from 21 January 2018 to 24 January 2018 with the collaboration of Mirko Surdi (Ghent University), Francesco Fassi and Fausta Fiorillo (3D Survey Group, Politecnico di Milano), and Catherine Mittermayer (University of Geneva). The survey documented with high-definition 3D models (photogrammetry and structured-light scanner) a small lot of cuneiform tables, foundation cones, and votive plaques of Mesopotamian origin from the Durac-Donner private collection.

4.  It is important to note that with regard to Dino-Lite USB microscopes, the reported magnification incorporates digital magnification forms . Thus, as suggested by the manufacturer, in this case it is often more useful to compare field of view rather than magnification, i.e., to compare the physical dimensions of the original item being magnified to the resulting size of the item on the display.

5.  The reference zero for the WD value is the outer end of the plastic nozzle; so, negative WD values are motivated by the need to make the specimen compenetrate the nozzle (assuming the specimen size allows this).

6.  Also known as photogrammetric principal distance.

7.  For this optical system, adjusting the magnification is equivalent to moving the lens plane backwards or forwards between the object and the sensor. In this mechanism, therefore, it is not possible to define for all magnifications a single value of $f$ (main distance), or of $H$ (flight height), or of their sum $D = f + H$, since the distance to the object will vary, as will the position of the center of the optical unit.

8.  Root mean square error (RMSE) is the standard deviation of the residuals (prediction errors). In other words, it is a measure of accuracy that allows for the measurement of the difference between the predicted and observed values. From a mathematical point of view, it is the square root of the average squared errors. Larger errors have a disproportionately large effect on RMSD; consequently, outliers affect RMSD. It is always non-negative, and a value of 0 (almost never achieved in practice) would indicate a perfect fit to the data.

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
