# Peer review of "Cuneiform Tablets Micro-Surveying in an Optimized Photogrammetric Configuration"

_heritage, doi:10.3390/heritage5040162_

Round 1
Reviewer 1 Report
Very good work about 3D documentation of tiny objects (cuneiform tablets in this paper) and very well presented.
Congrats to the authors for their work!!
Author Response
The authors sincerely thank the reviewer's congratulations.
Reviewer 2 Report
In this paper, the authors investigated remote sensing methodology for creating 3D representations of tiny artifacts. While the topic in this paper seems to be interesting for attracting attention in the field, the reviewer has a few comments attached which may help to improve the quality of the paper.
1) In the introduction section, it may be worth adding some discussions on why the 3D reconstruction of small artifacts is important. Although the authors spent some effort discussing the research gaps in the field, most of the discussions are unclear/incomplete. For example, the discussion on Page 2 contains few references. It is hard to gauge what has been done in the past.
2) The reviewer wishes to see the discussion of the proposed method against similar methods in the fields, particularly in these two studies. The authors mentioned one of them in the paper but it was not discussed in the literature review:
· Sapirstein, P. (2018). A high-precision photogrammetric recording system for small artifacts. Journal of Cultural Heritage, 31, 33-45.
· Marziali, S., & Dionisio, G. (2017). Photogrammetry and macro photography. The experience of the MUSINT II Project in the 3D digitizing process of small size archaeological artifacts. Studies in Digital Heritage, 1(2), 298-309.
3) With the aforementioned being said, it is unclear how the proposed method can bridge the research gaps and bring a new perspective to the researchers and practitioners.
4) Section 2.2 seems to be strange to be added to the paper as the discussion is quite lengthy and the topics here are not strongly related to the scope of the paper. The comparison of different methods could be published in a different venue. This writing breaks the flow of the paper in the reviewer's view.
5) Is Section 2.2.4 on 3D photogrammetry reconstruction the study to be performed in this paper? If so, why compare it against other methods upfront without showing the results?
6) The authors illustrate the aim in Section 3.1. Wouldn’t it be too late to mention it in the paper? The research aim shall be brought up upfront.
7) Sections 4 and 5 are abrupt and break the flow of the paper. The reviewer did not understand the purpose of these discussions. They seem to be under different topics. The reviewer wishes the authors could see the comments below for revising the paper.
8) It may be worth paying attention to the writing language during revision.
Overall, the paper was not well written with many topics packed together with no clear flow that allows the reader to follow. It may be wise to revisit the paper and make major changes to the structure of the paper. The reviewer likes to see a discussion on the research background, motivation, existing work and limitations, contribution, and research objectives in the introduction section. Then, it shall be followed by a brief review of the background and existing efforts of small artifacts in this study (3 pages at most; not 8 to 9 pages as currently illustrated in the paper). Next, the hardware setup along with parameters for image data acquisition shall be provided. If the authors like to discuss any technology features, this would be the place. Thereafter, results shall be shown neatly with a discussion on these results. Finally, it is the conclusion section.
Author Response
The authors thank the reviewers for the meticulous text analysis, which has allowed us to improve the article. Before promptly responding to the reviewers’ comments, the authors would like to note that all integrations in the text have been highlighted in orange. Instead, minor corrections were made using the review mode.

Reviewer 3 Report
The paper is devoted to the development of a technique for obtaining photographs for three-dimensional digitization of small artifacts based on the SfM principle, using a USB microscope.
First of all, the main part of the article (method and results) largely repeats the previously published work:
Antinozzi, S.; Fiorillo, F. Optimized Configurations for Micro-Photogrammetric Surveying Adaptable to Macro Optics and Digital Microscope. The International Archives of the Photogrammetry, Remote Sensing and Spatial Information Sciences 2022, XLVI-2/W1, pp. 25-32. https://doi.org/10.5194/isprs-archives-XLVI-2-W1-2022-25-2022.
The correctness of the re-publication of these materials will be left to the discretion of the editors.
In addition, the article has a significant number of issues that need to be corrected.
1. In the introduction, there are very few references to works on solving 3D digitization issues using macroscale photogrammetry, both in general and in cultural heritage in particular (for example, works by Gajski, D.; Angheluta, L.M.; Plisson, H; Vavulin , M.; Rodríguez-Martín, M. and others). The theme is well developed and it needs to be shown.
2. The introduction mentions active sensors and their high cost. Examples should be given with specific models and their cost (or just links).
3. The authors of one of the key features of their work indicate the low cost of the system. But is it really so? To confirm these words, approximate prices should be indicated. How much cheaper is the microscope used than a macro lens or magnifying rings? Why is commercial and not free software used if the goal of the work is to develop a low-cost system?
4. The introduction says that "Precisely for this reason, part of the research focused on configuring a high-performance hardware setup system designed for both: i) the most established solutions of macro-photogrammetry, based on digital cameras combined with macro lenses; ii) and also portable digital microscopes, whose validity has been tested for microscale surveying also in our previous works." - but in the article there is nothing at all about cameras with macro lenses, only a usb microscope.
5. In the introduction, in part "ii) and also portable digital microscopes, whose validity has been tested for microscale surveying also in our previous works", references should be made to the relevant works.
6. Section 2 is very well written - clear, concise, interesting. There is only one note here, the authors need to study the works of Hubert Mara devoted to improving the display and recognition of characters on three-dimensional models of cuneiform tablets, and insert links in section 2.2.4.
7. Section 4 describes attempts to restore the optical system of the microscope, and it also says that in the end "the manufacturer was asked what type of sensor is mounted on this system". These recovery attempts are not needed in the article. A block with a brief description of the optical system and technical parameters of the microscope can be placed at the beginning of section 3, which describes the equipment used.
8. In section 5 "In short, the process is streamlined and speeded up as we start from close to actual data." - experimental data that would confirm these words are not presented.
9. The results are not supported by studies. It is said that the use of a microscope proved to be very effective, but at the same time, the article does not compare resolution, accuracy, data acquisition time, data processing time or solution cost for solutions with a USB microscope, camera with a macro lens and 3d scanner. What criteria were used to measure "effectiveness"?
10. Also in the article there is nothing about “The various experimental survey campaigns made it possible to define the most suitable and fastest purchase acquisition setup in the function of the object shape and the required detail”
11. Figure 25 has nothing to do with text.
In general, the article has a low scientific value, since the possibilities of macro-photogrammetry have been worked out quite widely. However, it remains relevant, as it demonstrates the possibility and quality of 3D digitization using a USB microscope.
Author Response

(The authors gave the same response as above.)

Reviewer 4 Report
Review for paper heritage-1895751
This is a good example of quality scientific work. The main contribution of the work is defining the most suitable hardware configuration and acquisition method for obtaining a digital copy of a very small object with a complex and detailed surface such as cuneiform tablets. Photogrammetric recording was proposed and a custom configuration for a digital microscope was created and such a solution was very effective. Several experiments were created in order to arrive at the optimal solution, which includes fast data acquisition, high-resolution model and the presence of texture. Also, the cost of this procedure is low and it can be used as an algorithm in cultural heritage institutions. Finally, I think that it is a very interesting and good paper I suggest the Editorial Board to accept it for publication.
Author Response
The authors sincerely thank the reviewer's encouragement and positive comments
Round 2
Reviewer 2 Report
I recommend the paper for publication.
Author Response
The authors thank you for taking the time to the paper.
Reviewer 3 Report
The quality of the article has improved significantly after restructuring, adding relevant literature and making other changes.
An error was also noticed in specifying the scale in Fig. 18. It must be corrected before publication.
Author Response
The authors thank you for taking the time to the paper.
The error in figure 18 was fixed.